

# A decade-plus of Antarctic sea ice thickness and volume estimates from CryoSat-2 using a physical model and waveform-fitting

Steven Fons[1,2], Nathan Kurtz[2], and Marco Bagnardi[2,3]

[1]Earth System Science Interdisciplinary Center, University of Maryland, College Park, MD, USA
[2]Cryospheric Sciences Laboratory, NASA Goddard Space Flight Center, Greenbelt, MD, USA
[3]ADNET Systems, Inc., Bethesda, MD, USA

**Correspondence:** Steven Fons (steven.w.fons@nasa.gov)

**Abstract.** We utilize a physical waveform model and a waveform-fitting method to estimate the snow depth and snow freeboard of Antarctic sea ice from CryoSat-2, and use these estimates to calculate the sea ice thickness and volume over an 11+ year time series. We compare our snow depth and thickness estimates to other altimetry- and ship-based observations, and find good agreement overall with some discrepancies in certain regions and seasons. The time series is used to calculate trends in the data, and we find small but statistically significant negative trends in the Ross Sea autumn (-0.3 cm yr$^{-1}$), the Eastern Weddell winter (-0.8 cm yr$^{-1}$), and the Western Weddell autumn and annual-average (-2.6 and -1.6 cm yr$^{-1}$, respectively). Significant positive trends are found in the pan-Antarctic summer (0.4 cm yr$^{-1}$) and Amundsen-Bellingshausen winter and annual-average (2.3 and 0.9 cm yr$^{-1}$, respectively). Though pan-Antarctic trends in sea ice thickness and volume are small between 2010-2021, we find larger-magnitude trends regionally and since 2014. We place these thickness estimates in the context of a longer-term, snow-freeboard-derived, laser-radar sea ice thickness time series that began with ICESat and continues with ICESat-2. Reconciling and validating this longer-term, multi-sensor time series will be important in better understanding changes in the Antarctic sea ice cover.

## 1 Introduction

Sea ice thickness is an important parameter in Earth's climate system as it controls fluxes of heat, moisture, and salinity between the ocean and atmosphere (Persson and Vihma, 2016). It also acts as an indicator of climate change and variability (EPA, 2016) due to its intimate relationship with other components of the cryosphere. Knowledge of sea ice thickness has long been important to indigenous Arctic communities and continues to be a focus today for climate studies as well as maritime navigation in icy waters (Bourke and Garrett, 1987; De Silva et al., 2015; Holland et al., 2006). While measuring thickness in situ is a straightforward process - requiring only a hole and a measuring device - the sheer size of the sea ice pack in both hemispheres and the impracticality of routine work in the polar regions limits the ability to manually measure the thickness of the sea ice cover on the basin-scale. Instead, satellite altimeters are typically used.

By measuring the height of the sea ice above the local sea surface (i.e. the freeboard) from altimetry, one can apply assumptions of hydrostatic balance to estimate the thickness of the sea ice. Wavelength differences between the two primary types of altimeters - radar and laser - correspond to varying dominant scattering horizons over sea ice and therefore different





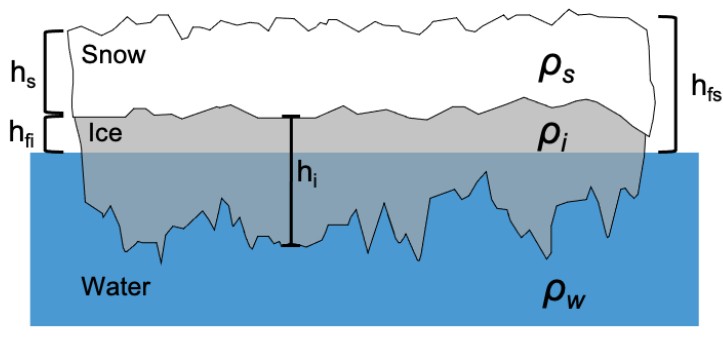

**Figure 1.** Schematic showing parameters in Equations 1 and 2, including snow depth ($h_s$), ice freeboard ($h_{fi}$), snow freeboard ($h_{fs}$), ice thickness ($h_i$), and the density terms of snow ($\rho_s$), ice ($\rho_i$), and seawater ($\rho_w$).

retrieved freeboards: ice freeboard (assuming the dominant scattering comes from the snow-ice interface) and snow freeboard, respectively. These two distinct freeboard values require different equations for calculating thickness. If the ice freeboard is known, one can define thickness as:

$$h_{i-ifb} = \left( \frac{\rho_w}{\rho_w - \rho_i} \right) h_{fi} + \left( \frac{\rho_s}{\rho_w - \rho_i} \right) h_s, \tag{1}$$

where $h_{i-ifb}$ is the ice thickness computed from the ice freeboard, $h_{fi}$ is the ice freeboard, $h_s$ is the snow depth, and $\rho$ is the
density of seawater ($\rho_w$), ice ($\rho_i$), and snow ($\rho_s$). If the snow freeboard is known, then the equation instead becomes:

$$h_{i-sfb} = \left( \frac{\rho_w}{\rho_w - \rho_i} \right) h_{fs} + \left( \frac{\rho_s - \rho_w}{\rho_w - \rho_i} \right) h_s, \tag{2}$$

where $h_{i-sfb}$ is the ice thickness computed from the snow freeboard, $h_{fs}$ is the snow freeboard, and the snow depth and density are as in Equation 1 (Kurtz and Markus, 2012; Kwok, 2011). It is clear from Equations 1 and 2 that estimates of freeboard and snow depth are the two necessary measurements for deriving thickness from satellite altimetry.

Over Arctic sea ice, satellite-altimeter-derived freeboard has been estimated since around 2003 using the ERS-1 and ERS-2 satellites (Laxon et al., 2003). The launch of NASA's ICESat in 2003 facilitated sea ice freeboard measurements between 2003-2008 (Zwally et al., 2002; Kwok et al., 2007; Kurtz et al., 2008; Farrell et al., 2009), while ESA's Cryosat-2 satellite has been used extensively to estimate Arctic sea ice freeboard since its launch in 2010 (Laxon et al., 2013; Kurtz et al., 2014; Ricker et al., 2014; Kwok and Cunningham, 2015; Tilling et al., 2018; Landy et al., 2020). In most of these cases, a regional
snow depth climatology built from ground-based measurements collected between 1954-1991 is used to convert the freeboard measurements to thickness estimates (Warren et al., 1999). Other studies have employed lower-resolution snow depth data from passive microwave sensors (Kurtz et al., 2009). More recently, NASA's ICESat-2 satellite has been operating, and studies have combined ICESat-2 freeboards with snow depths from models (e.g. Petty et al. 2018, Liston et al. 2018, Liston et al. 2020) to estimate sea ice thickness (Petty et al., 2020).



While the studies mentioned above have found success in estimating sea ice freeboard over Arctic sea ice, fewer works have done so for Antarctic sea ice. This hemispheric discrepancy is primarily due to two reasons. First, ice freeboard from radar altimetry is difficult to estimate for Antarctic sea ice, since the thicker snow layer on Antarctic sea ice can depress the ice nearer to the ocean surface, leading to flooding, enhanced brine wicking, and other processes that drive complex stratigraphy and complicate returns from Ku-band altimeters (Giovinetto et al., 1992; Maksym and Jeffries, 2000; Willatt et al., 2010).

Ground-based studies have shown that assuming a Ku-band radar pulse penetrates to the snow-ice interface is likely incorrect, with dominant returns more likely to originate from within the snow layer or the air-snow interface (Willatt et al., 2010). Second, the snow depth distribution over Antarctic sea ice is not as well known (Giles et al., 2008). Unlike the Arctic, few ground-based snow depth estimates exist from which thickness can be estimated. Worby et al. (2008) provides seasonal and regional snow depth estimates from ship-based observations, however, these are subject to seasonal and spatial sampling biases

due to the location of ship tracks. Other studies have used passive microwave instruments to derive snow depth, however, these measurements contain large pixel-level uncertainties associated with ice type, sensor footprint size, and a lack of validation data (Markus and Cavalieri, 2013; Kern and Ozsoy-Çiçek, 2016; Maksym and Markus, 2008). These barriers mean that using radar altimetry, mainly CryoSat-2, over Antarctic sea ice will increase complexity of and uncertainty in the freeboard retrievals. Also, it means that calculating sea ice thickness will require either assumptions about the snow layer or new basin-wide datasets

of snow depth on sea ice.

        Some works have attempted to retrieve Antarctic sea ice freeboard from CryoSat-2 (Paul et al., 2018; Schwegmann et al., 2016; Fons and Kurtz, 2019). Schwegmann et al. (2016) estimated radar freeboard, but the lack of snow depth information prevented the correction for wave speed in snow (in order to convert to ice freeboard) and the estimation of thickness. Paul et al. (2018) did correct for wave speed using passive microwave snow depths, but did not estimate thickness from the measurements,

citing the need for external snow depth data. Fons and Kurtz (2019) estimated snow freeboard from CryoSat-2 using snow scattering information contained in a physical waveform model, but also did not estimate thickness citing a lack of confidence in the snow depth retrievals. Currently, the only Antarctic sea ice thickness products from CryoSat-2 come from the ESA Climate Change Initiative (CCI, Hendricks et al. 2018), who utilized the procedure from Schwegmann et al. (2016) and Paul et al. (2018) in their estimates, and from Garnier et al. (2022), who calibrated Envisat thicknesses with CryoSat-2 to produce

a longer time series. However, due to the uncertainties surrounding the radar penetration, snow depth, and the use of power-threshold empirical retracking algorithms, these estimates are both assumed to be biased high (Hendricks et al., 2018; Garnier et al., 2022).

        Other studies have been able to estimate Antarctic sea ice thickness from laser altimetry - namely ICESat and ICESat-2 - through the use of key snow depth assumptions. Kurtz and Markus (2012) applied a "zero ice freeboard" (ZIF) assumption,

which assumes that the snow depth depresses the ice down to the water level everywhere, resulting in a snow depth equal to the snow freeboard and an ice freeboard equal to zero. Under this assumption, Equation 2 becomes:

$$h_{i-0ifb} = \left( \frac{\rho_s}{\rho_w - \rho_i} \right) h_{fs}. \tag{3}$$





Zero ice freeboards have been observed during ground-based measurement campaigns (Willatt et al., 2010), however, this assumption applied basin-wide likely underestimates the actual ice thickness (Kwok and Kacimi, 2018; Kacimi and Kwok, 2020). Other studies have applied a static ratio between freeboard and snow depth derived from passive microwave measurements and assumed a single ice/snow layer in their ICESat retrievals (Kern et al., 2016; Li et al., 2018; Xu et al., 2021; Wang et al., 2022). Xu et al. (2021) improved the one-layer method to be able to estimate thickness from ICESat-2 alone. In each of these studies, however, the lack of reliable basin-scale snow depth information added uncertainty and limited confidence in the thickness retrievals.

Since the launch of ICESat-2 in 2018, two cryosphere-focused satellite altimeters have been observing sea ice in simultaneous operation (ICESat-2 and CryoSat-2). The contrasting instrument wavelengths provide new potential to estimate snow depth on sea ice by differencing the freeboards retrieved by each sensor, $h_{fs} - h_{fi}$. Kwok et al. (2020) first showcased this on Arctic sea ice, showing good agreement with snowfall patterns from reanalysis. In the Antarctic, Kacimi and Kwok (2020) used the same technique to derive six months of snow depth on Antarctic sea ice, and used the resulting snow depths along with the retrieved freeboards to provide estimates of sea ice thickness and volume. Their results showed physically-realistic values that provide useful insight into the Antarctic sea ice thickness distribution, however, they rely on the assumption that Ku-band pulses originates from the snow-ice interface. Additionally, their work is restricted to only time in which both satellites are operating and uses only near-coincident data (time difference <10 days) to estimate snow depth.

While the ICESat and ICESat-2-based studies of Antarctic sea ice provide a good baseline into Antarctic sea ice thickness, they only cover the years 2003-2008 and late 2018 onward. Clearly, there is a large need to utilize CryoSat-2 to fill in the gap between these two measurement periods (Meredith et al., 2019). Additionally, the need to estimate snow depth utilizing large assumptions or external coincident data constrains the current retrievals in terms of confidence as well as temporal coverage.

To address the above needs, this study utilizes a CryoSat-2 waveform-fitting method to estimate the physical properties of Antarctic sea ice and generate an 11-year record of Antarctic sea ice thickness from CryoSat-2. This method, hereafter referred to as the CryoSat-2 Waveform-Fitting method for Antarctic sea ice (CS2WFA), relies on a forward waveform model and optimization procedure to assist in retrieving the elevation of both the air-snow and snow-ice interfaces over Antarctic sea ice. Knowledge of these two interface locations enables the estimation of snow depth from the fitted CryoSat-2 waveform alone. While previous works have described the algorithm in detail (Fons and Kurtz, 2019) and assessed snow freeboard retrievals with independent data (Fons et al., 2021), this work showcases retrievals of snow depth on sea ice over the entire CryoSat-2 mission, and combines these estimates with snow freeboard data to estimate the sea ice thickness and volume from 2010-2021.

## 2 Data

### 2.1 CryoSat-2

The primary data used in this work come from ESA's CryoSat-2 satellite, which launched in April 2010. CryoSat-2's primary payload is SIRAL, a synthetic aperture radar (SAR) altimeter operating in the Ku-band at around 13.6 GHz (Wingham et al., 2006). This work utilizes Baseline-D Level 1-B waveform data from both SAR and SARIn modes around the Antarctic con-



tinent (European Space Agency, 2019a, b). These data come from footprints that are approximately 1.65 km across-track and 380 m along track, though impacts from off-nadir leads can originate from over 10 km away (Tilling et al., 2018; European Space Agency, 2019a). Baseline-D processing covers the time period from 16 July 2010 until 21 August 2021 with small gaps due to data availability throughout (European Space Agency, 2019c). This entire time range is used in this study. To ensure

consistency, no Baseline-E data (which began 22 August 2021) is included in this work, since back-processed Baseline-E data were not available at the time of submission.

In this study, each CryoSat-2 file is processed following the methods outlined in Section 3. Focus is given to pan-Antarctic retrievals, so all data are gridded to the NSIDC 25 km polar stereographic grid (EPSG: 3976) to generate monthly and seasonal means.

**2.2 Ancillary data**

Monthly snow freeboard climatology maps from ICESat and ICESat-2 are used as an initialization for this retrieval process (described in Section 3.1). These are monthly mean maps of sea ice snow freeboard (12 in total) consisting of an average of ICESat data from 2003-2008 and ICESat-2 data from 2018-2019 (described in Fons et al. (2021)). These data are averaged from ICESat freeboards from Kurtz and Markus (2012) and ICESat-2 ATL10 freeboards (Kwok et al., 2021). Each monthly

map is used for initializing the corresponding month of CryoSat-2 data, regardless of the year.

All monthly maps are restricted to grid cells that contain at least 50% ice concentration. To distinguish between lower concentrations, the Bootstrap Version 3 sea ice concentration algorithm is used (Comiso, 2017). This algorithm is based off of brightness temperatures from Nimbus-7 SMMR and DMSP SSM/I and SSMIS instruments, and is provided as daily and monthly averages. The monthly averages are utilized in this study. These data are also used to compute the areal coverage of

the sea ice for a given month, where grid cells with concentrations greater than or equal to 50% are used in the area calculation. Area is calculated simply by multiplying the sea ice concentration present in each grid cell by the area of the grid cell, and summing the entire grid or region of interest.

Finally, other datasets are used as comparisons to the retrievals shown. Kacimi and Kwok (2020) (KK20) used freeboard data collected by CryoSat-2 and ICESat-2 over Antarctic sea ice to estimate snow depth and thickness. They showed results

from the year 2019 as monthly regional means and standard deviations of each parameter. Results from KK20 are given here to help validate these CryoSat-2 retrievals. Additionally, two datasets are used here as a measure of comparison against regional thickness values. One is the ship-based observations of sea ice thickness from the Antarctic Sea Ice Processes and Climate (ASPeCt) program (Worby et al., 2008). These estimates were compiled from thousands of observations around the continent, spanning over two decades (1981 - 2005). The other regional comparison dataset comes from Xu et al. (2021), who estimated

sea ice thickness from ICESat-2 between 2018-2020 using an improved one-layer method (OLMi).



## 3 Methods

This section describes the procedure for retrieving sea ice thickness from CryoSat-2 using CS2WFA. The procedure is broken down into the elevation retrieval (Section 3.1), the freeboard and snow depth estimation (Section 3.2), the calculation of thickness and volume (Section 3.3), and the estimate of uncertainties (Section 3.4).

### 3.1 Waveform-fitting and elevation retrieval

The primary methodology for retrieving sea ice properties used in this work is the CryoSat-2 waveform-fitting retrieval algorithm put forth in Fons and Kurtz (2019) and improved in Fons et al. (2021). This process is a physical retracking method that employs a forward waveform model to track the sea ice surfaces on the waveform, which allows for the retrieval of sea ice elevation. The retrieval algorithm is described in detail in Fons and Kurtz (2019), however, a broad overview is given in this

section.

First, CryoSat-2 level 1B data are ingested and each individual waveform is classified into its respective surface-type: floe-type (originating from sea ice), lead-type (originating from open water cracks in the ice), ocean-type (originating from the open ocean around the ice pack), and mixed-type (an ambiguous return from mixed-surfaces). This is done by analyzing the pulse peakiness, stack standard deviation, and skewness of each waveform (discussed in Fons et al. (2021)). Ocean-type and

mixed-type waveforms are discarded prior to processing.

After classifying the returns, a physically-modeled waveform is constructed for each individual CryoSat-2 echo. The model is given, generally, by the equation:

$$\Psi(\tau) = P_t(\tau) \otimes I(\tau, \alpha) \otimes p(\tau, \sigma) \otimes \upsilon(\tau, h_{sd}) \tag{4}$$

where $\tau$ is the echo delay time relative to the time of scattering from the mean scattering surface and $\otimes$ represents a convolution

of the compressed transmit pulse, $P_t(\tau)$, the rough surface impulse response, $I(\tau, \alpha)$, the surface height probability density function, $p(\tau, \sigma)$, and the scattering cross section per unit volume, $\upsilon(\tau, h_{sd})$ (Brown, 1977; Kurtz et al., 2014; Fons and Kurtz, 2019). For floe-type waveforms, this model is fed with four different initial parameters: snow depth, snow-ice interface tracking point, surface roughness, and angular backscattering efficiency (further defined in Table 1). These parameters follow those given in Fons et al. (2021), with the exception that the amplitude scale factor was found to have negligible impact on the results

and was removed to reduce model complexity. This model assumes a fixed ratio between the air-snow and snow-ice interface backscatter, with the snow-ice interface backscatter being 6 dB greater than the air-snow interface (Kwok, 2014). Lead-type waveforms have no snow cover (by definition), and therefore the $\upsilon$ term goes to a delta function at $\tau = 0$, resulting in three free parameters: snow-ice interface tracking point (which is simply the ice surface tracking point), roughness, and angular backscattering efficiency. These parameters are derived from waveform characteristics, when applicable, or otherwise from

independent datasets (Table 1).

The next step involves fitting the modeled waveform to the actual CryoSat-2 waveform through a bounded trust-region Newton least-squares optimization approach. Bounds are provided for each input parameter about the initial guess to constrain



**Table 1.** Free parameters used in the CS2WFA retrieval algorithm and static parameters used in the volume scattering term of the waveform model. Additional static parameters used can be found in Fons and Kurtz (2019) and Kurtz et al. (2014).

| Free Parameters | | Initial Value | Bounds | Reference |
|---|---|---|---|---|
| $h_s$ | Snow depth | ICESat/ICESat-2 monthly "climatology" | $\pm$ 30 cm | Kurtz and Markus (2012); Kwok et al. (2021) |
| $t$ | Snow-ice interface time delay | 70% power threshold | $\pm$ 3 ns | Laxon et al. (2013) |
| $\sigma$ | Roughness (std of surface height) | 0.15 m | 0-1 m | Fons and Kurtz (2019) |
| $\alpha$ | Angular backscatter | Lookup table based on waveform characteristics | 1.5e1-9e8 | Kurtz et al. (2014) |
| Static Parameters | | | | |
| $\sigma^0_{sfc-snow}$ | Snow surface backscatter | 0 dB | - | Arthern et al. (2001) |
| $\sigma^0_{sfc-ice}$ | Ice surface backscatter | 6 dB | - | Kwok (2014) |
| $\sigma^0_{vol-snow}$ | Snow volume backscatter | -7 dB | - | Beaven et al. (1995) |
| $\sigma^0_{vol-ice}$ | Ice volume backscatter | -17 dB | - | Beaven et al. (1995) |
| $K_{e-snow}$ | Snow extinction coefficient | 0.1 m$^{-1}$ | - | Ulaby et al. (1982) |
| $K_{e-ice}$ | Ice extinction coefficient | 5 m$^{-1}$ | - | Ulaby et al. (1982) |

the optimization to best-guess physically-realistic values for a given location or waveform shape (Table 1). Each function evaluation adjusts the input parameters within the provided bounds, until a minimum residual between the modeled and actual waveforms is found, or until a maximum number of function evaluations (100) is reached. The output "fit parameters" provide estimates of the actual values of each parameter in the waveform model. The large assumptions are that (a) this waveform model accurately can represent a CryoSat-2 return from the sea ice surface and (b) the fitting procedure, when initialized with physically-realistic inputs and bounds, can find the global minimum optimization result as opposed to getting caught in a local minimum. Fit parameters are discarded if the result is a "poor fit", which is described as having a resnorm (squared norm of the residual) greater than 0.3.

The output parameters of snow-ice interface tracking point and snow depth provide (when accounting for wave speed through the snowpack) the locations of the physical air-snow and snow-ice interfaces on the waveform as a function of radar return time. Retracking corrections are then calculated that relate these locations to the nominal tracking bin provided in the CryoSat-2 data product. To retrieve the surface elevation ($h_e$), the retracking correction is added to a provided geophysical correction and the raw CryoSat-2 range. This retrieval is shown by:

$$h_e = A - R_0 \tag{5}$$

where $A$ is the altitude of the satellite center of gravity above the World Geodetic System 1984 (WGS84) ellipsoid and $R_n$ is the range from the satellite to the surface, given by:

$$R_0 = R_n + c_r + c_g \tag{6}$$



where $R_n$ is the raw range computed from the time delay to a reference point of the range window, $c_r$ is the retracking correction, and $c_g$ is the geophysical correction. Equation 5 results in the elevation of the surface above the WGS84 ellipsoid. These elevations can then be used to estimate freeboard and snow depth.

It must be reiterated that the estimation of ice freeboard and snow depth requires an estimate of the snow-ice interface elevation. As has been previously stated, empirical threshold retrackers typically assume the dominant scattering horizon is located at the snow-ice interface (and that the snow layer is mostly transparent at Ku-band frequencies), which can lead to an overestimation of the ice freeboard and is a primary reason behind low confidence in retrievals of ice freeboard over Antarctic sea ice (Paul et al., 2018; Schwegmann et al., 2016). This physical retracking method does not contain the same assumption and instead uses the physical model and fitting method to estimate the actual snow-ice interface by accounting for attenuation of the signal in snow and ice layers. Therefore, it is hypothesized that this physical method better tracks that snow-ice interface compared to empirical approaches, which motivates the calculation of ice freeboard and snow depth. This idea is explored further in subsequent sections.

## 3.2 Estimating freeboard and snow depth

Once the snow depth and snow-ice interface are retrieved using the retracking procedure described above, the air-snow interface elevation and freeboard can be estimated. The air-snow interface elevation is found simply by adding the output snow depth to the snow-ice interface elevation. To estimate freeboard, the first step is determining the sea surface height (SSH). The elevation of all lead-type waveforms are averaged in 10 km along-track segments and linearly interpolated, following Kwok et al. (2022). Lead segments are only calculated if at least three lead-type points exist within. This along-track SSH is then subtracted from every floe-type elevation, yielding a freeboard estimate for each floe-type point. Subtracting the SSH from the air-snow interface elevation results in a snow freeboard estimate, while that from the snow-ice interface results in an ice freeboard estimate. The output snow depth parameter gives our estimate of the snow thickness. The effects of the snow layer on radar wave speed propagation are accounted for in the snow-ice interface retracking and the ice freeboard calculation, following Mallett et al. (2020) and shown in Fons et al. (2021).

Each floe-type waveform with a good fit, therefore, has a snow freeboard, ice freeboard, and snow depth estimate, provided that a co-located 10 km sea surface height segment also exists. Estimates of freeboard and snow depth shown here are gridded onto the NSIDC 25 km polar stereographic grid.

## 3.3 Calculating sea ice thickness and volume

After obtaining estimates of sea ice freeboard and snow depth on sea ice, the thickness of the sea ice can be calculated by applying the hydrostatic assumption. This assumption follows that an object (in this case, snow-covered sea ice) immersed in a fluid will be buoyed with a force equal to that due to gravity. Combining the estimates of the freeboard and snow depth with estimates for the density of seawater, snow, and sea ice (as in Equations 1 and 2), one can calculate the thickness.

Typical values of sea ice and snow density used for Antarctic sea ice thickness calculation vary depending on the study. Most studies tend to use single, static density values across all seasons; ice density values tend to be around 915 to 917 kg





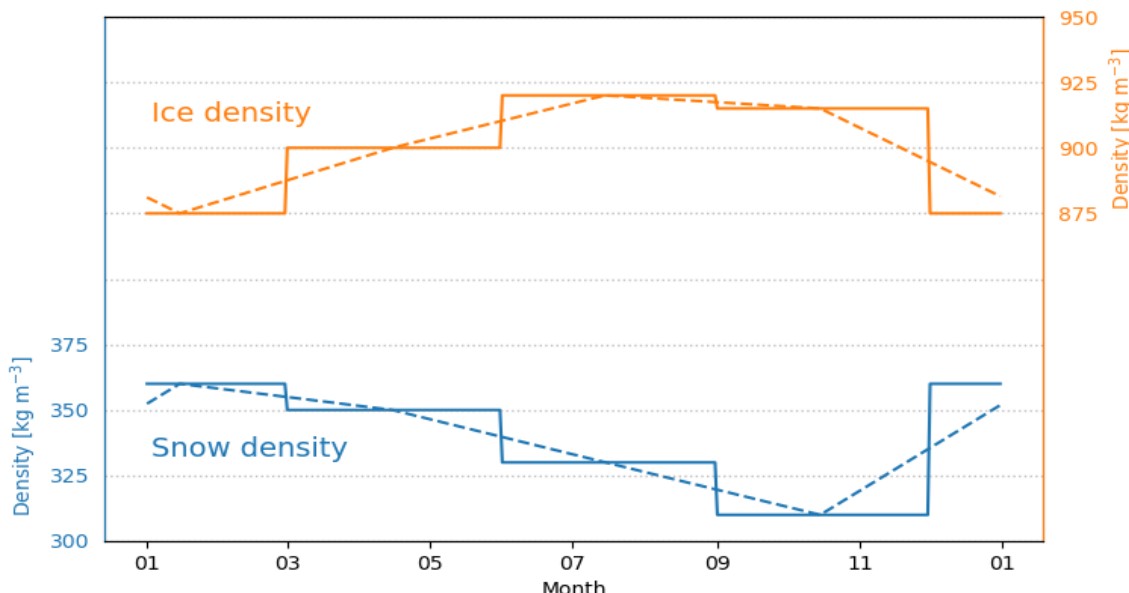

**Figure 2.** Sea ice (orange) and snow (blue) density values used in the thickness calculation. Solid lines are the fixed seasonal density values, while the dashed line is the daily linear interpolation used in the thickness calculation.

m$^{-3}$, while snow densities used tend to be 300 to 320 kg m$^{-3}$ (Kern et al., 2016; Li et al., 2018; Kacimi and Kwok, 2020). Kurtz and Markus (2012) varied these values seasonally, based off in situ measurements collected around the continent. For ice
density, they used 900, 875, and 900 kg m$^{-3}$ for spring, summer, and autumn, respectively, based off the measurements collated in Worby et al. (2008) and Buynitskiy (1967). For snow density, they used 320, 350, and 340 kg m$^{-3}$ for spring, summer, and autumn, respectively, following Massom et al. (2001).

In this work, we use seasonally-varying snow and ice density values, but interpolate them through time as an attempt to better capture a seasonal signal that is likely present as the sea ice evolves. To do this, we provide a "seasonal value" of density that
is assigned to the approximate midpoint day of each season: 15 January for summer, 15 April for autumn, 15 July for winter, and 15 October for spring. For ice density, these seasonal values follow Kurtz and Markus (2012), Hutchings et al. (2015), and Buynitskiy (1967), and are set to 875 (summer), 900 (autumn), 920 (winter), and 915 (spring) kg m$^{-3}$. The snow density values are set to 360 (summer), 350 (autumn), 330 (winter), and 310 (spring) kg m$^{-3}$, following Massom et al. (2001) and Kurtz and Markus (2012). Then, these seasonal values are linearly interpolated between the midpoint dates, providing daily density
estimates that are used in the thickness calculation. The interpolated densities are shown in Fig. 2. For the density of seawater, we use a static value of 1024 kg m$^{-3}$.

Like freeboard and snow depth, thickness is also gridded using the NSIDC polar stereographic grid. Mean values shown herein are pan-Antarctic or regional averages, and do not take the sea ice area into account (i.e. results shown are not area-





weighted means). Volume is computed by multiplying basin-average sea ice thickness by the areal coverage of sea ice. The sea
ice area is computed using the Bootstrap sea ice concentration data, where each 25 km grid cell concentration above 50% is
multiplied by the area of the grid cell. Then, this total area value can be multiplied by the basin- or region-averaged thickness
to compute the volume.

For the freeboard, snow depth, thickness, and volume results given herein, both monthly- and seasonally-averaged data are
shown both pan-Antarctic as well as from individual regions. The seasons are broken up as follows: (1) summer, D-J-F (2)
autumn, M-A-M (3) winter, J-J-A (4) spring, S-O-N. The regions are longitudinally demarcated as follows: (1) Ross Sea,
$160° - 230°$ (2) Amundsen-Bellingshausen Seas (Am-Bel), $230° - 300°$ (3) Western Weddell Sea, $300° - 315°$ (4) Eastern
Weddell Sea, $315° - 20°$ (5) Indian Ocean, $20° - 90°$ (6) Pacific Ocean, $90° - 160°$. These regions are shown in Fig. 3.

### 3.4 Estimating uncertainty in sea ice thickness and volume retrievals

In this section, an estimate of the uncertainty in the thickness and volume measurements are provided through a Gaussian
error propagation method (following Spreen et al. 2009; Kern and Spreen 2015; Petty et al. 2020). The total uncertainty in a
thickness measurement comes from a combination of random and systematic uncertainties. Random uncertainties in thickness
measurements can be calculated (Petty et al., 2020) however, because they are random, we assume they decrease substantially
when many individual thickness measurements are averaged together. Since thickness results in this study are shown only over
large scales and averaged monthly to a 25 km grid, we assume the random uncertainty becomes negligible basin-wide and
do not explicitly calculate it here. Future work involving along-track thickness estimates and validation would necessitate a
calculation of random uncertainty as well as an estimate of the uncertainty brought on by the fitting procedure itself.

Instead, an estimate of the systematic uncertainty at the grid-cell-scale is given here. We assume these uncertainties are
correlated and represent bias in the measurement, and therefore cannot be reduced through averaging (Ricker et al., 2014).
Here, the systematic uncertainty is estimated following Petty et al. (2020) where:

$$\sigma^2_{hi-sfb} = \sigma^2_{hs}\left(\frac{\rho_s}{(\rho_w - \rho_i)} - \frac{\rho_w}{(\rho_w - \rho_i)}\right)^2 + \sigma^2_{\rho s}\left(\frac{h_s}{(\rho_w - \rho_i)}\right)^2 + \sigma^2_{\rho i}\left(\frac{h_{fs}\rho_w}{(\rho_w - \rho_i)^2} + \frac{h_s\rho_s}{(\rho_w - \rho_i)^2} - \frac{h_s\rho_w}{(\rho_w - \rho_i)^2}\right)^2. \quad (7)$$

In Equation 7, $\sigma_{hi-sfb}$ is the uncertainty in sea ice thickness estimated using snow freeboard (Eqn. 2), $h_s$ is the snow depth,
$h_f s$ is the snow freeboard, and $s$, $w$, $i$ represent the density ($\rho$) and uncertainty ($\sigma$) of snow, seawater, and sea ice, respectively.
The snow depth uncertainty $\sigma_{hs}$ is taken as the standard deviation of snow depth measurements with each 25 km grid cell,
following Petty et al. (2020). Densities of snow ($\rho_s$), ice ($\rho_i$), and water ($\rho_w$) are set to the values given in Kurtz and Markus
(2012), with the uncertainty in the snow density ($\sigma_{\rho s}$) taken as 50 kg m$^{-3}$ and the uncertainty in the ice density ($\sigma_{\rho i}$) taken as
20 kg m$^{-3}$. The uncertainty in the seawater density is assumed to be negligible and is not included in this calculation (Kurtz and
Markus, 2012; Kern et al., 2016; Li et al., 2018). Monthly average 25 km grids of snow depth ($h_s$) and snow freeboard ($h_f$)
are used in the uncertainty calculation, and therefore uncertainty estimates are provided for each grid cell basin-wide. Monthly
average thickness uncertainty values range from around 22 to 45 cm, and are given (basin-wide) in Table 2.

An estimate of the volume uncertainty is also provided, but done so in a simplistic way compared to other methods over
Arctic sea ice (Tilling et al., 2018). Here, a Gaussian error propagation approach is used to combine the uncertainty due to sea



**Table 2.** Monthly pan-Antarctic sea ice thickness ($h_i$) and volume ($V$) uncertainty found in this study, shown both as absolute ($\sigma$) and fractional ($\delta$) values. All values are 2010 - 2021 averages.

| Month | $\sigma_{h_i}[m]$ | $\delta h_i/h_i[\%]$ | $\sigma_V[km^3]$ | $\delta V/V[\%]$ |
|-------|------|------|------|------|
| Jan. | 0.22 | 18 | 569 | 18 |
| Feb. | 0.26 | 20 | 464 | 21 |
| Mar. | 0.32 | 33 | 967 | 33 |
| Apr. | 0.38 | 37 | 2398 | 38 |
| May | 0.39 | 36 | 3890 | 37 |
| Jun. | 0.42 | 38 | 5636 | 39 |
| Jul. | 0.45 | 40 | 8088 | 40 |
| Aug. | 0.44 | 38 | 8066 | 39 |
| Sep. | 0.42 | 37 | 8333 | 37 |
| Oct. | 0.38 | 33 | 6596 | 33 |
| Nov. | 0.29 | 26 | 4893 | 27 |
| Dec. | 0.24 | 21 | 1638 | 21 |

ice thickness and sea ice area, the two components that make up the volume calculation. This is provided as an initial volume uncertainty estimate until more information on the uncertainty relating to snow depth and snow/ice density from Antarctic sea ice is known.

This estimated volume uncertainty is given by:

$$\frac{\delta V}{|V|} = \sqrt{\left(\frac{\delta h_i}{h_i}\right)^2 + \left(\frac{\delta A}{A}\right)^2}, \tag{8}$$

where $\delta V/|V|$ is the monthly average sea ice volume fractional uncertainty, $\delta h_i/h_i$ is the sea ice thickness fractional uncertainty, and $\delta A/A$ is the fractional uncertainty in sea ice area. In this study, a conservative fractional uncertainty value for sea ice area of 5% is used, following Spreen et al. (2009). The fractional uncertainty in thickness ($\delta h_i/h_i$ ) is found by dividing the

thickness uncertainty grid by the mean thickness grid and computing the basin-wide average. The right-hand side of Equation 8 provides a fractional combined uncertainty in sea ice volume. Multiplying this fractional uncertainty with the mean volume estimate yields a quantitative sea ice volume uncertainty. This uncertainty ranges generally between 18-40% of the total sea ice volume (Table 2) and is dominated by the uncertainty in thickness. Once again, these values are simply first estimates used to constrain the volume estimates in the context of the observed interannual variability. They are provided, pan-Antarctic, in

Table 2.



## 4 Results

The methodology described in Section 3 was applied to all CryoSat-2 data collected over the Southern Ocean from the beginning of the mission in July 2010 until August 2021, resulting in over 11 years of data. This time period is hereafter referred to as the "CryoSat-2 period". In this section, we show results from CS2WFA, comparisons with other datasets, and observed trends in the data. The following sections cover snow depth (Section 4.1), thickness (Section 4.2) and volume (Section 4.3).

### 4.1 Snow depth on sea ice

The retrieved snow-ice interface elevations are subtracted from the air-snow interface elevation estimates, providing along-track snow depth on sea ice for the duration of the CryoSat-2 period. This sections showcases a snow depth comparison as well as seasonal patterns in the snow depth distribution.

The lack of reliable in situ data involving snow depth on Antarctic sea ice poses a challenge when attempting to validate snow depth retrievals from CryoSat-2. Here, snow depth estimates from KK20 are used as a point of comparison for region-scale snow depth retrievals from CryoSat-2. Figure 3 shows a regional comparison between CS2WFA and KK20 snow depths for the year 2019. Data from KK20 are only provided in April-November and are missing in July due to a gap in data collection from ICESat-2. Differences are generally within 5 cm, though can be larger in regions where snow (and ice) thickness is typically greatest, especially the Western Weddell sector. In other regions, KK20 mean snow depths almost always fall within the CS2WFA IQR. The two datasets exhibit a similar seasonal pattern, however, KK20 snow depths exhibit less growth over the year, as the change from ∼June to November is generally smaller than that from CS2WFA.

Monthly differences in pan-Antarctic snow depth (CS2WFA minus KK20) range from 0.2 cm in April to 7.5 cm in November. There are some caveats, however. While these measurements both use CryoSat-2 data to estimate the snow-ice interface elevation, KK20 also uses ICESat-2 data and only collects snow depths within +/- 10 days from a valid ICESat-2 freeboard measurement. Therefore, it must be noted that each month in this comparison involves different amounts of data between the two methods which could explain some of the differences observed. This is especially true in November, when KK20 only uses the first two weeks of ICESat-2 data, while these CS2WFA data come from the whole month. Despite some of these differences, the observed similar seasonality and small mean differences in the snow depths between both datasets is nevertheless encouraging. However, more validation data - especially in Austral summer - would be useful to better assess the snow depths retrieved from both of these methods.

Figure 3 also gives insight into the summer snow depths retrieved from CS2WFA. In all sectors except the Western Weddell, the thickest average snow depths are found to occur in summer months (usually February). This finding is consistent with what is found in Worby et al. (2008), and is hypothesized to be due to the fact that the older ice near the coast that survives summer melt has had more time for snow accumulation and therefore thicker snow depths. Additionally, the regions with the largest difference in snow depth in summer compared to other months (i.e. Pacific, Indian, Ross) experience the largest ice extent change between winter and summer, which could explain these substantial differences in the regional averages.





**Figure 3.** Comparison of the CS2WFA-retrieved snow depth to that from KK20 for each region in the year 2019. Boxplots are from the CS2WFA data for each month, where the boxes show the inter-quartile range (IQR), whiskers show 1.5 times the IQR, horizontal lines are the medians, and purple dots are the mean values. Green points and whiskers represent the mean value and standard deviation from KK20. In the pan-Antarctic panel, the means are simply an average of all the individual region means, and therefore do not include standard deviations.





In the Western Weddell, some of the thinnest average snow depths occur in the late summer (F-M). This finding is seemingly counter-intuitive given the behavior of the other regions, however, was also observed by Worby et al. (2008). One possible explanation could be due to the presence of surface melting of the snow layer that can act to reduce snow depths and/or potentially impact retrievals in this region and season. Markus and Cavalieri (2013) found that surface melt in summer months in this region can occur - due in part to the more-northern location compared to other regions - and found that this melt can complicate passive microwave returns. Here, it is possible that surface melt could lead not just to thinner snow depths, but also could lead to erroneous CryoSat-2 waveform classification, as melt-affected floe-points may appear more specular in radar returns. This could cause mis-classification of sea ice returns, which could impact the SSH and retrieved snow-ice interface elevations, potentially leading to anomalous snow depths. Another explanation is that the yearly change in ice area in the Western Weddell is smaller than in any other region. This consistency in ice extent means that the region-averaged seasonal cycle would be controlled more so by actual changes in the snow depth (e.g. from surface melt or accumulation) than by changes in the ice extent that get averaged into the regional mean (e.g. from new ice formation that has near-zero snow).

The 2010-2021 mean snow depth on Antarctic sea ice for each season is given in Fig. 4, along with the basin-wide snow depth distribution for all grid cells between 2010 and 2021. Most notably, these spatial patterns and distributions appear more realistic (based off of other estimates from Markus and Cavalieri 2013, and Kacimi and Kwok 2020) than what was attempted in Fons and Kurtz (2019), signalling that the model and retrieval improvements made in Fons et al. (2021) had a substantial - and beneficial - effect on the snow depth retrievals. In Fig. 4, one can see the seasonal growth in snow thickness, as any given region shows an increase from autumn to winter to spring. Also, the distributions show the impact of ice extent on the basin-averaged snow depth, as they shift from broad (little extent but thicker snow) to thin (large extent with thinner snow on average). Pan-Antarctic snow depths range between 16 and 25 cm on average, with the thickest basin-averaged snow depths in summer (consistent with the comparisons shown in Fig. 3). Standard deviations of snow depth range between 9.4 and 12.9 cm.

Overall, the retrieved CS2WFA snow depths show reasonable spatial and temporal distributions that generally compare with estimates from KK20, despite the sampling differences. In the next section, these snow depths are combined with snow freeboard estimates (detailed in Fons et al. 2021) to estimate sea ice thickness.

## 4.2 Sea ice thickness

The retrieved snow depths shown above are used in Equation 2 to calculate the sea ice thickness. Here, estimates of Antarctic sea ice thickness derived from CryoSat-2 from the years 2010 - 2021 are shown.

The monthly mean Antarctic sea ice thickness distributions and spatial patterns from 2010 to 2021 are given in Fig. 5. Basin-wide thicknesses are largest in February (1.26 m), when ice extent is at a minimum, and lowest in March (0.95 m), when new ice has began to form. Monthly standard deviations range from 59 to 78 cm. The summer distributions tend to be broader than other months, though all months exhibit a singular mode. The spatial pattern follows closely with that of snow depth (Fig. 4, with the largest thicknesses being found in the Western Weddell sector, as well as along the coast in the Am-Bel and Pacific sectors, while thinner ice tends to be found away from the coast in the Eastern Weddell, Indian, and Ross sectors. These values tend to be thicker than those observed in previous studies (e.g. Worby et al. 2008), which is discussed later in this section.





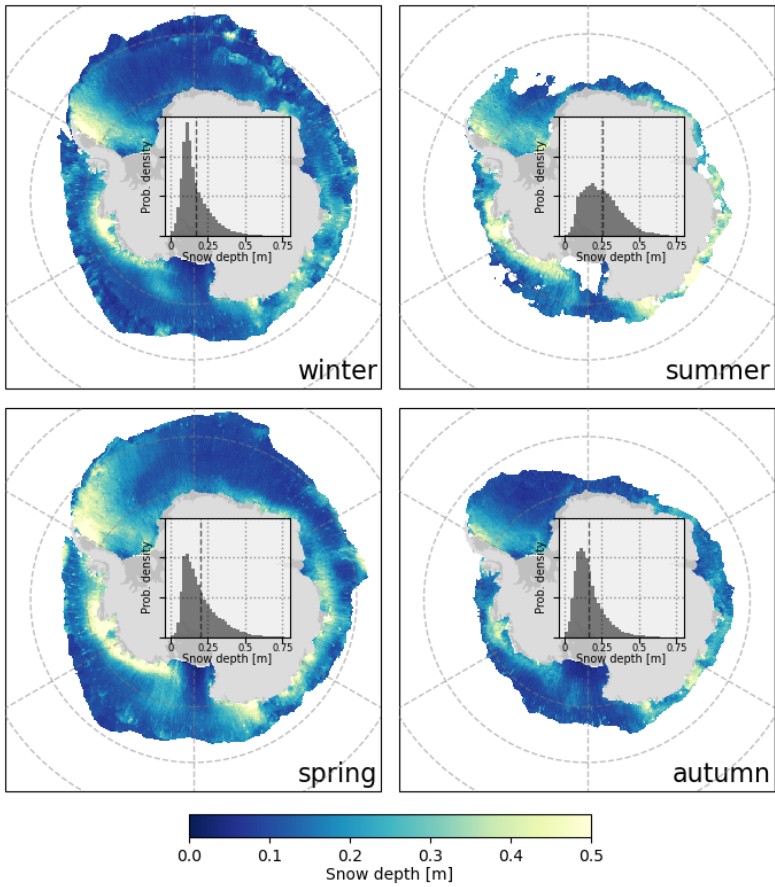

**Figure 4.** Seasonal average (July 2010 - August 2021) maps and distributions of snow depth from CS2WFA. Histogram bin sizes are 2 cm.

When averaged over all years, the monthly thickness distributions in Fig. 5 tend to mask some of the variability present both regionally and seasonally. To investigate this variability, Fig. 6 gives thickness distributions broken down by region and season. The regional variability is apparent, as the Pacific, Western Weddell, and Am-Bel sectors showcase broader distributions (density $\lessgtr 1$) with thicker mean values, while the other regions show a narrower distribution (probability density $\gtrless 1$) and slightly thinner means. The largest seasonality within a region is found in the Pacific sector, where the summer ice thickness is noticeably thicker than the other seasons, owing to the low extent present during this season. Like what was found in the snow depth results, the Western Weddell summer shows the thinnest ice thickness of all seasons. This result once again matches what was found in Worby et al. (2008), and is likely due in part to the relatively stable seasonal ice extent in this region that shows the basal melt occurring in summer and contributes less to a "thickening" by way of averaging over a low ice extent.

Crosses and triangles in Fig. 6 on the bottom and top show the seasonal/region mean values given in Worby et al. (2008) and Xu et al. (2021), respectively. In most regions, these CS2WFA thicknesses tend to estimate thicker ice than Worby et al.





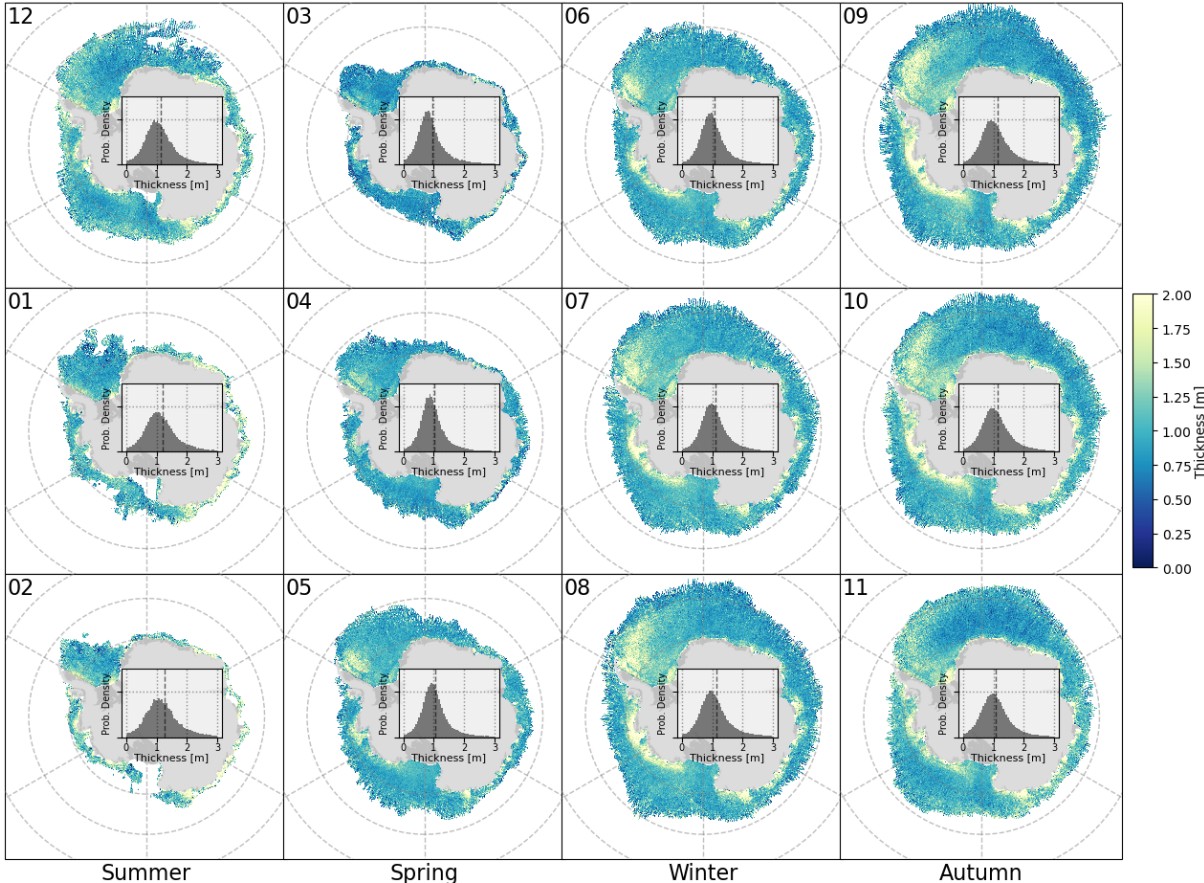

**Figure 5.** Monthly average (July 2010 - August 2021) maps and distributions of sea ice thickness from CS2WFA, arranged in columns by season. Histogram bin sizes are 5 cm.

(2008), but are generally comparable to Xu et al. (2021). Despite some differences between the datasets, a few similarities can be seen. First, mean thicknesses in the Western Weddell sector agree well between all three datasets. Additionally, Worby
et al. (2008) found the thickest sea ice to be in summer in the Ross, Am-Bel, Indian, and Pacific Ocean sectors. This CryoSat-2 method also finds these regions to be the thickest in summer (with the exception of the Ross Sea being within one cm of the thickest season), while Xu et al. (2021) finds three of these regions to be thickest in summer. Both Worby et al. (2008) and Xu et al. (2021) find summer to be the thinnest season in the Western Weddell sector, matching what we show with CS2WFA.

Another interesting feature in the seasonal distributions is the perceived lack of seasonal variability in some regions. Some
sectors showcase more seasonal variation in the distributions, while others show a very similar mode and distribution shape in all seasons. This lack of seasonality contrasts what is found in Arctic sea ice, where a clear seasonal cycle exists in the thickness distributions (Petty et al., 2020). One contributing factor is the fact that these distributions are created from the entire



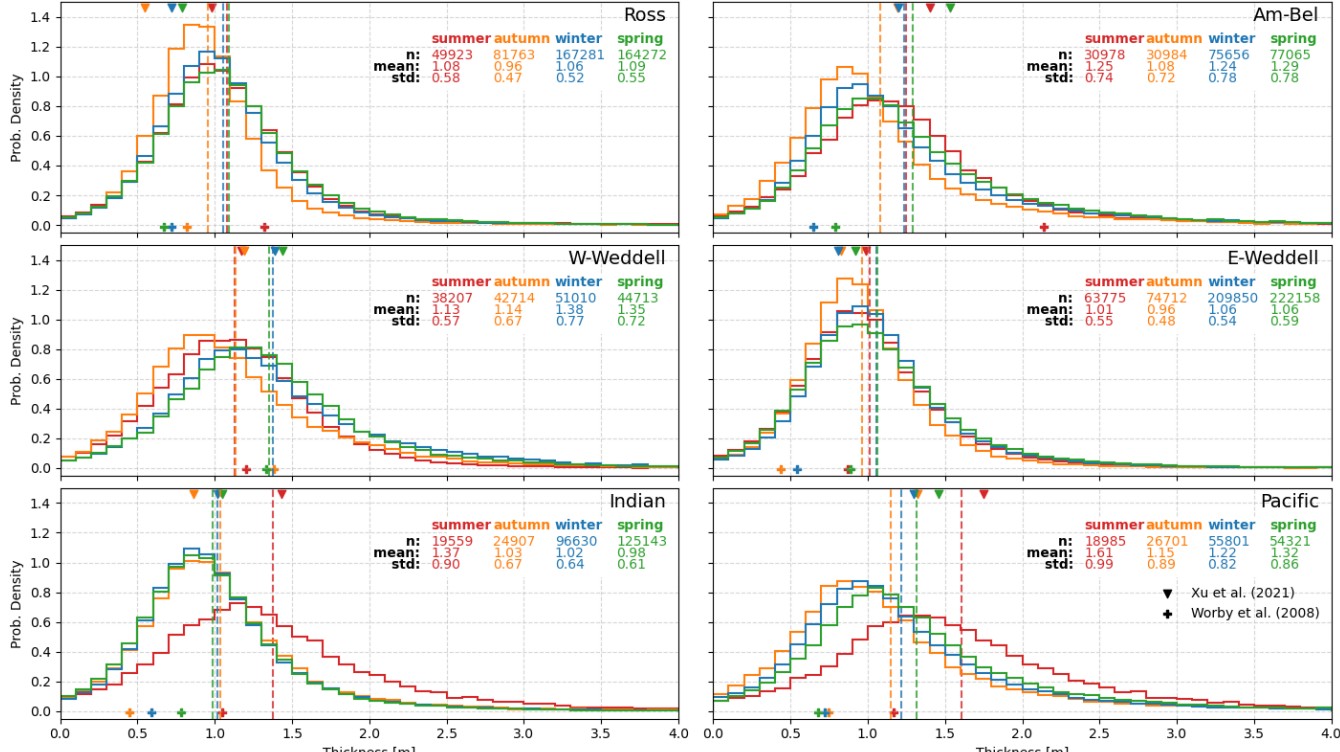

**Figure 6.** Seasonal thickness distributions from each of the six regions covering 2010-2021. Red represents summer, orange represents autumn, blue represents winter, and green represents spring, where the vertical dashed line gives the mean value. The number of grid cells, mean, and standard deviations of the distributions are given. Bin sizes are 10 cm. Crosses on the bottom and triangles on the top mark the seasonal and regional mean values from Worby et al. (2008) and Xu et al. (2021), respectively.

11-year record. When looking at an individual region and season over time (Fig. 7), one can see much larger variability in the regional and seasonal mean values.

Figure 7 shows the pan-Antarctic sea ice thickness (solid black line) averaged as a 31-day rolling mean, as well as from individual sectors (panels) from July 2010 until August 2021. When viewed in this sense, a clear seasonal cycle of thickness emerges: pan-Antarctic averaged thickness increases through Austral summer, reaching a maximum around February with the minimum in sea ice extent (Parkinson, 2019). The pan-Antarctic average thickness then falls in March when new ice forms. After, there is a gradual thickening through autumn and winter as both ice extent and basal growth continues. This thickening

slows and switches to a slight thinning by the middle and end of spring, as basal growth slows and melt begins basin-wide. The thickness only increases on average once the thinnest ice melts entirely in the beginning of summer, leaving only the thickest ice present near the continent. This cycle continues each year.

While the time series in Fig. 7 shows the general seasonal cycle described above, there does exist some variability in the magnitude of the mean thickness. For example, the February peaks in ice thickness vary between less than 1.2 and more that





**Figure 7.** Time series of Antarctic sea ice thickness covering the entire CryoSat-2 period 2010-2021. The black line represents the pan-Antarctic average, and is shown by itself and with all individual regions (panels). The shaded regions show the estimated uncertainty pan-Antarctic and in each individual region.

1.3 m on average, while the March minima also vary between ∼0.85 and 1.05 m. This variation can be much larger amongst individual regions.





Compared to other studies, the mean thickness values shown here appear to be slightly higher than expected, especially in months of new ice growth where one would expect much of the ice cover to be below 1 m. Kurtz and Markus (2012) found mean values below 1 m for all seasons, with the caveat of using the ZIF assumption. Worby et al. (2008) found thinner

annual and seasonal means from ship-based observations (Fig. 6), while Maksym and Markus (2008) showed thinner passive microwave-based thicknesses. However, other satellite-based studies found thicknesses comparable - and even thicker than - these CS2WFA results. As shown in Fig. 6, Xu et al. (2021) found regional and seasonal thickness values comparable to these from CryoSat-2. KK20 acknowledged that their estimates seem high, and included multiple estimates of thickness to "correct" for the anomalously-thick results, introducing correction factors of $\delta = 3$ cm and $\delta = 6$ cm. These CS2WFA

thickness estimates are slightly thinner that what was estimates by KK20, and closer to their $\delta = 6$ cm values. While it is possible that these satellite-based methods are too high – potentially from anomalous snow-ice interface tracking, surface type mis-characterization, or other factors – it is also possible that the Antarctic sea ice thickness distribution is simply thicker than previously assumed (Williams et al., 2015) and that ship-based observations and ZIF satellite estimates simply underestimate the actual thickness. More discussion on the "actual" thickness of Antarctic sea ice is provided in Section 5.

**4.3 Sea ice volume**

For each month of sea ice thickness data shown above, the monthly mean sea ice volume is also computed. This is done by multiplying the sea ice areal coverage in the basin or region by the mean ice thickness. A time series of volume is given in Fig. 8, where the Pan-Antarctic total volume (black line) is broken down into contributions from the various sectors. The sea ice volume varies substantially within a year, ranging from around 2500 km$^3$ to around 20000 km$^3$. The Ross and Eastern Weddell

sectors contribute the largest percentage of volume in the winter and fall seasons (when area is greatest), while the Western Weddell sector tends to contribute most to the spring and summer pan-Antarctic sea ice volume. These values compare overall to the volume estimates put forth in KK20, both in the basin-wide sense as well as in the individual regions.

The time series of volume in Fig. 8 follows closely with the time series in sea ice extent shown in Parkinson (2019), which makes sense given that the pan-Antarctic sea ice volume calculation is dominated in magnitude by the sea ice area.

This similarity is evident during the volume maximum in 2014 and the volume minimum in 2017, which correspond to the largest and smallest Antarctic sea ice extents, respectively, found over the satellite record (Parkinson, 2019). There is clearly more interannual variability in months of greater sea ice volume (August - October) than in months of less volume (January - March).

**4.4 Trends in sea ice thickness and volume**

Given this 11-year record of Antarctic sea ice thickness and volume, we investigate potential trends in the data. While the time series in Figs. 7 and 8 do not immediately show a clear trend in thickness nor volume over the CryoSat-2 period, a seasonal and regional look into trends paints a clearer picture.

Figure 9 shows seasonal trend maps that give the change in thickness between 2010-2021 for each 25 km grid cell basin-wide. The slope of the regression line of each grid cell time series is calculated using the Theil-Sen estimator, and is only





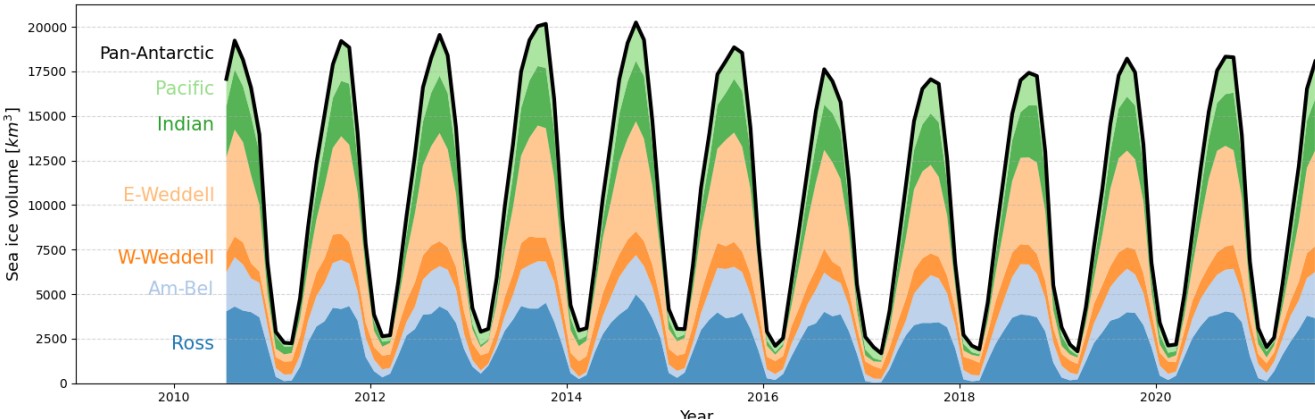

**Figure 8.** Time series of Antarctic sea ice volume covering the entire CryoSat-2 period 2010-2021. The dark black line represents the pan-Antarctic total volume, shown as the sum of the individual regions.

reported if a respective grid cell contains at least four years of data. To reduce noise in the trend data, sea ice thickness maps are first smoothed with a 5x5 grid cell Gaussian kernel filter, which corresponds to an effective size of 125x125 km (following Kurtz and Markus 2012). The region-average Theil-Sen slopes are given in Table 3. To determine the statistical significance of the reported trends, the p-values and z-values are reported, where the p-values are from the Mann-Kendall Test of the null hypothesis that there is no monotonic trend in the time series, and the z-values indicate the normalized Mann-Kendall score, where negative values indicated a negative trend and vice versa. Bold values in Table 3 reject the null hypothesis that there is

no trend in the data, and are statistically significant at a level better than 95%.

One can initially see lots of variation in the trends around the continent in all seasons, however, some areas exhibit larger magnitude trends. In autumn, most of the Western Weddell sector exhibits a strong thinning, with a region mean of over 2.6 cm $yr^{-1}$, which is larger than the trends observed here in other seasons (1 to 1.6 cm $yr^{-1}$ in summer and winter and 0.3 in spring). In

winter, there is a large area of thinning predominantly in the Eastern Weddell Sea, corresponding to the statistically significant regional trend observed of -0.8 cm $yr^{-1}$ on average (Table 3). There is also an area of thickening in the Am-Bel region (2.3 cm $yr^{-1}$), which agrees with the observation-based trends in Garnier et al. (2022) and modeled trends shown in Holland (2014) for this region. In summer, each region shows a slight positive trend on average, with the exception of the Indian and W. Weddell sectors, which show thinning of 2.1 and 1.1 cm $yr^{-1}$, respectively. This contrasts Xu et al. (2021), who found summer thinning

in all regions, albeit covering different time range and using different instruments. The Pan-Antarctic trend in the summer is 0.4 cm $yr^{-1}$ and is statistically significant (Table 3). When considering annual-average trends, the W. Weddell region shows a statistically significant thinning of around 1.6 cm $yr^{-1}$, while the Am-Bel region shows a significant thickening of 0.9 cm $yr^{-1}$. Both of these trends agree with Garnier et al. (2022), who found similar negative and positive trends in these respective regions.





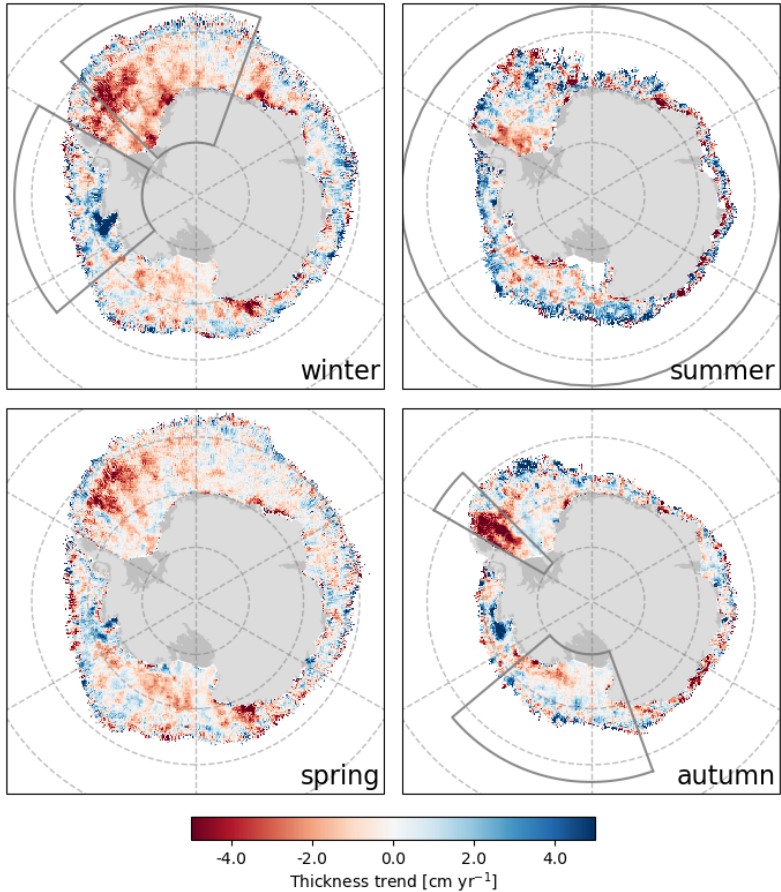

**Figure 9.** Seasonal trends in Antarctic sea ice thickness between 2010-2021. Red values indicate thinning, while blue values indicate thickening over time. Grey outlines show regions that demonstrate statistically significant trends as shown in Table 3.

## 5 Discussion

### 5.1 What do the observed trends tell us?

When considering the entire Antarctic sea ice pack, the observed trends are quite small over the CryoSat-2 period. This is especially noteworthy when considering that trends in Arctic sea ice thickness and volume are around an order of magnitude greater than what is shown here (Kwok and Cunningham, 2015). Given their small pan-Antarctic magnitudes, what can one make of these trends in the context of the earth system?

For one, the pan-Antarctic trends highlight the need to consider the Antarctic ocean-ice-atmosphere system in a regional context. These small Pan-Antarctic trends are the product of larger magnitude, competing regional trends that tend to compensate in all regions and seasons. The competing regional trends not only showcase that is there a strong regional heterogeneity





**Table 3.** Antarctic sea ice thickness trends from 2010 - 2021 broken down by region and season. Trends give the slope of the regression line found using the Theil-Sen estimator. The p- and z-values of the reported trends are also given, where p-values come from the Mann-Kendall Trend Test and z-values are the normalized Mann-Kendall Score. Bold value indicate trends statistically significant at a level better than 95%.

| Season | | Ross | Am-Bel | W. Weddell | Sector E. Weddell | Indian | Pacific | Pan-Antarctic |
|---|---|---|---|---|---|---|---|---|
| Summer | Ice thickness trend (cm yr⁻¹) | 0.70 | 0.85 | -1.06 | 0.91 | -2.12 | 0.85 | **0.43** |
| | p (z) value | .28 (1.09) | .28 (1.09) | .09 (-1.71) | .16 (1.40) | .21 (-1.25) | .28 (1.09) | **.04 (2.02)** |
| | Ice volume trend (km³ yr⁻¹) | -28.1 | 12.6 | -16.4 | -68.4 | -5.1 | -12.7 | -98.9 |
| | p (z) value | .16 (-1.40) | .12 (1.56) | .35 (-0.93) | .21 (-1.25) | .99 (0.00) | .44 (-0.78) | .21 (-1.25) |
| Autumn | Ice thickness trend (cm yr⁻¹) | **-0.27** | 0.38 | **-2.63** | -0.01 | -0.02 | -0.11 | -0.35 |
| | p (z) value | **.04 (-2.02)** | .53 (0.62) | **.04 (-2.02)** | .99 (0.00) | .88 (-0.16) | .88 (-0.16) | .12 (-1.56) |
| | Ice volume trend (km³ yr⁻¹) | -17.8 | 20.1 | **-29.4** | -64.8 | -10.3 | -9.9 | -87.9 |
| | p (z) value | .76 (-0.31) | .21 (1.25) | **.02 (-2.34)** | .16 (-1.40) | .35 (-0.93) | .35 (-0.93) | .35 (-0.93) |
| Winter | Ice thickness trend (cm yr⁻¹) | -0.35 | **2.26** | -1.61 | **-0.77** | 0.12 | -0.43 | -0.29 |
| | p (z) value | .16 (-1.40) | **.04 (2.02)** | .16 (-1.40) | **.03 (-2.18)** | .53 (0.62) | .64 (-0.47) | .06 (-1.87) |
| | Ice volume trend (km³ yr⁻¹) | -36.4 | -7.8 | **-23.1** | -35.6 | -19.3 | -19.4 | -140.9 |
| | p (z) value | .06 (-1.87) | .76 (-0.31) | **.01 (-2.65)** | .35 (-0.93) | .21 (-1.25) | .06 (-1.87) | .16 (-1.40) |
| Spring | Ice thickness trend (cm yr⁻¹) | -0.15 | 1.61 | -0.36 | -0.75 | -0.25 | 0.13 | -0.41 |
| | p (z) value | .53 (-0.62) | .28 (1.09) | .76 (-0.31) | .16 (-1.40) | .64 (-0.47) | .88 (0.16) | .53 (-0.62) |
| | Ice volume trend (km³ yr⁻¹) | -35.5 | 2.3 | -3.2 | -22.1 | -33.4 | -17.2 | -78.1 |
| | p (z) value | .16 (-1.40) | .88 (0.16) | .76 (-0.31) | .64 (-0.47) | .12 (-1.56) | .64 (-0.47) | .44 (-0.78) |
| Annual | Ice thickness trend (cm yr⁻¹) | -0.18 | **0.91** | **-1.59** | -0.31 | -0.21 | 0.03 | -0.12 |
| | p (z) value | .99 (0.00) | **.03 (2.15)** | **.03 (-2.15)** | .21 (-1.25) | .37 (-0.89) | .72 (0.36) | .11 (-1.61) |
| | Ice volume trend (km³ yr⁻¹) | -53.8 | -0.3 | -19.8 | -62.1 | -22.8 | -19.4 | -175.8 |
| | p (z) value | .15 (-1.43) | .99 (0.00) | .07 (-1.79) | .37 (-0.89) | .21 (-1.25) | .15 (-1.43) | .21 (-1.25) |

in the sea ice pack in terms of its physical properties (such as snow depth and thickness), but also likely point to regional discrepancies in the ocean and atmospheric processes that drive changes in sea ice. For example, Holland (2014) found regional
differences in modeled dynamic processes (e.g. wind stress) around the continent, with the Am-Bel region showing large trends in dynamic forcings that likely contribute to the positive trends in sea ice thickness observed there. This regional perspective is crucial when conducting sea ice and earth system studies in the Southern Ocean.

    Another important outcome of these observed trends in Antarctic sea ice – especially in the context of a warming climate – is the need to assess trends over different timescales. When viewed over the entire CryoSat-2 period, the observed pan-Antarctic
trends are small in magnitude. However, the same trends over a shorter, more recent time period (2014 - 2021) show mostly larger trends, with some regions showing more than two times greater magnitudes (Table 4). In the Pacific region, the annual-average trend flips from very slight positive over all years to a negative trend since 2014. Similar increases in the trends of sea ice thickness and volume since 2014 are also found in Garnier et al. (2022), including the switch from thickening to thinning in the Pacific sector. This recent "speed-up" in the trends in Pan-Antarctic thickness and volume parallels the recent changes
in sea ice extent over the same time period, where Parkinson (2019) found declines since 2014 at rates exceeding those seen in



**Table 4.** As in Table 3 but for the years 2014-2021.

| Season | | Ross | Am-Bel | W. Weddell | E. Weddell | Indian | Pacific | Pan-Antarctic |
|---|---|---|---|---|---|---|---|---|
| | | | | | Sector | | | |
| Summer | Ice thickness trend (cm yr⁻¹) | -0.63 | -0.07 | -1.12 | 0.98 | -1.74 | -0.19 | 0.39 |
| | p (z) value | .90 (-0.12) | .90 (-0.12) | .27 (-1.11) | .54 (0.62) | .27 (-1.11) | .71 (-0.37) | .39 (0.87) |
| | Ice volume trend (km³ yr⁻¹) | -51.7 | 30.1 | -33.2 | -130.6 | 29.8 | **-36.2** | -221.7 |
| | p (z) value | .27 (-1.11) | .06 (1.86) | .11 (-1.61) | .17 (-1.36) | .54 (0.62) | **.04 (-2.10)** | .17 (-1.36) |
| Autumn | Ice thickness trend (cm yr⁻¹) | -0.13 | 0.78 | **-3.98** | -0.83 | -0.11 | -1.69 | -0.59 |
| | p (z) value | .27 (-1.11) | .54 (0.62) | **.04 (-2.10)** | .06 (-1.86) | .71 (-0.37) | .17 (-1.36) | .11 (-1.61) |
| | Ice volume trend (km³ yr⁻¹) | -23.2 | 30.4 | **-50.6** | -162.7 | -6.9 | **-23.4** | -228.0 |
| | p (z) value | .90 (-0.12) | .11 (1.61) | **.01 (-2.60)** | .06 (-1.86) | .54 (-0.62) | **.04 (-2.10)** | .39 (-0.87) |
| Winter | Ice thickness trend (cm yr⁻¹) | -0.46 | **3.63** | -1.68 | -1.14 | -0.16 | -0.89 | -0.31 |
| | p (z) value | .39 (-0.87) | **.04 (2.10)** | .27 (-1.11) | .11 (-1.61) | .90 (-0.12) | .17 (-1.36) | .17 (-1.36) |
| | Ice volume trend (km³ yr⁻¹) | -44.7 | -14.9 | -16.4 | -104.1 | -12.5 | -18.4 | -169.8 |
| | p (z) value | .17 (-1.36) | .90 (-0.12) | .17 (-1.36) | .17 (-1.36) | .71 (-0.37) | .17 (-1.36) | .39 (-0.87) |
| Spring | Ice thickness trend (cm yr⁻¹) | -0.43 | 2.67 | -0.87 | -2.00 | -0.34 | 0.38 | -0.61 |
| | p (z) value | .76 (-0.30) | .37 (0.90) | .37 (-0.90) | .13 (-1.50) | .13 (-1.50) | .76 (0.30) | .55 (-0.60) |
| | Ice volume trend (km³ yr⁻¹) | -19.6 | 23.9 | -18.1 | -55.5 | 11.5 | 2.6 | -78.1 |
| | p (z) value | .99 (0.00) | .55 (0.60) | .76 (-0.30) | .99 (0.00) | .99 (0.00) | .99 (0.00) | .99 (0.00) |
| Annual | Ice thickness trend (cm yr⁻¹) | -0.34 | **1.17** | -2.75 | -0.71 | -0.21 | -0.34 | -0.18 |
| | p (z) value | .76 (-0.30) | **.04 (2.10)** | .13 (-1.50) | .13 (-1.50) | .76 (-0.30) | .37 (-0.90) | .23 (-1.20) |
| | Ice volume trend (km³ yr⁻¹) | -57.7 | 10.5 | -30.5 | -138.7 | -16.0 | -30.7 | -242.9 |
| | p (z) value | .37 (-0.90) | .76 (0.30) | .23 (-1.20) | .37 (-0.90) | .99 (0.00) | .13 (-1.50) | .55 (-0.60) |

the Arctic. Larger magnitude trends in recent years could signal that recent, rapid changes in the climate system (Arias et al., 2021) are impacting the Antarctic sea ice cover. Much like what was done in studies involving future projections of sea ice area (Bintanja et al., 2015), modeled studies into projected changes in Antarctic sea ice thickness and volume would be useful in better understanding how Antarctic sea ice may change under different climate scenarios.

Overall, the small magnitude trends in sea ice thickness and volume over the 2010 - 2021 time period makes it seem like the Antarctic sea cover is stable despite observed changes in the Southern Hemisphere system since 2010, such as an increase in the 0 - 2000 m ocean heat content, a decrease in the upper 100 m ocean salinity, and an increase in the near-surface atmospheric temperature (de Lavergne et al., 2014; Llovel and Terray, 2016; Jones et al., 2016). While it is possible that this stability signals a resilience by Antarctic sea ice to changes in the ocean-atmosphere system, more likely there are other contributing factors

– such as other atmospheric forcings or changes in Antarctic surface runoff – that help to balance the system and keep the Antarctic sea ice thickness relatively stable. Additionally, the larger magnitude trends seen regionally and in more recent years suggest that the Antarctic sea ice cover is not immune to change. A longer-term time series would be useful to better attribute trends in Antarctic sea ice thickness and volume and the possible climate implications.



## 5.2 Toward a reconciled laser-radar altimetry thickness record

With an 11+ year time series of Antarctic sea ice thickness from CS2WFA, one can start connecting the snow-freeboard-derived satellite-altimetry-based record of thickness that began with ICESat from 2003 - 2008 (Kurtz and Markus, 2012) and continues with ICESat-2 (2018-present). Figure 10 shows a full time series with thickness estimates from all three instruments (blue from ICESat, black from CryoSat-2, and green from ICESat-2). Here, we attempt to constrain the "actual" thickness of Antarctic sea ice by highlighting the range in estimates based on the technique used.

In Fig. 10, the solid lines represent thickness estimates made using the snow depths from CS2WFA, and can be thought of as our "best-estimate" thickness for the given period ($h_i$). For CS2WFA and ICESat-2 thicknesses, the monthly average snow depth grids are used in the calculation, while for ICESat a monthly snow depth climatology derived from these CS2WFA snow depths are used. Mean values are between ~0.7 and 1.5 m for CryoSat-2 and ICESat-2, while ICESat shows thicknesses between ~0.9 and ~1.8 m on average. The shaded region about the CS2WFA thickness time series represents our estimated 485 thickness uncertainty as calculated in Section 3.4. While the solid black line represents the best estimate from CS2WFA, the shaded region provides and upper and lower bound on what we estimate the "actual" thickness to be. It is encouraging that the ICESat-2 thicknesses – despite showing differences from CS2WFA – fall within the range of uncertainty.

The dashed lines in Fig. 10 indicate thicknesses computed using the ZIF assumption ($h_{i-ZIF}$). It is clear that using the ZIF assumption results in a thinner ice cover, with mean values always less than one meter. Likely, $h_{i-ZIF}$ is an underestimate 490 of the actual thickness, given that the ZIF assumption is not always valid in all regions and seasons (Kwok and Kacimi, 2018; Kacimi and Kwok, 2020). However, $h_{i-ZIF}$ follows closely along the lower bound of the CS2WFA uncertainty range, suggesting that it can be considered as a lower constraint on the actual sea ice thickness.

On the contrary, the dotted line in Fig. 10 shows the thickness estimated using a 70 % threshold retracker on the CryoSat-2 data ($h_{i-70}$). These values assume the 70 % power level on the CryoSat-2 waveforms gives the snow-ice interface elevation (and 495 thus ice freeboard), estimates the snow depth by subtracting from the waveform-fitting-retrieved air-snow interface elevation, and estimates thickness using this snow depth and freeboard. It is apparent that $h_{i-70}$ values are much thicker than other estimates, with mean values around 2 m on average. Likely, these values are an overestimate of the actual thickness, given what is known about radar penetration into the snow cover over Antarctic sea ice and how 70 % retrackers have been found to overestimate the snow-ice interface elevation (Schwegmann et al., 2016; Paul et al., 2018; Hendricks et al., 2018; Kacimi and 500 Kwok, 2020). The fact that the $h_{i-70}$ estimates fall well outside of the CS2WFA uncertainty range suggests that using a simple 70 % threshold retracker alone is not suitable to reliably estimate Antarctic sea ice thickness.

When viewed in this context, some dissimilarities between the thickness estimates and data sources emerge that should be the work of future study. For one, CS2WFA data appear to show small interannual variability, especially in $h_{i-ZIF}$, while the ICESat thickness tends to vary more year-to-year. This is especially apparent in the ICESat $h_i$, where the range is much 505 larger than is seen in CryoSat-2 and ICESat-2, and could come from the fact that ICESat was operating in discrete 'campaigns' and experienced large changes in laser energy over its lifetime. There is a substantial discrepancy in $h_i$ from CS2WFA and ICESat-2 in autumn 2019 and 2020. Given that the snow depths used are identical, these differences arise due to differences in





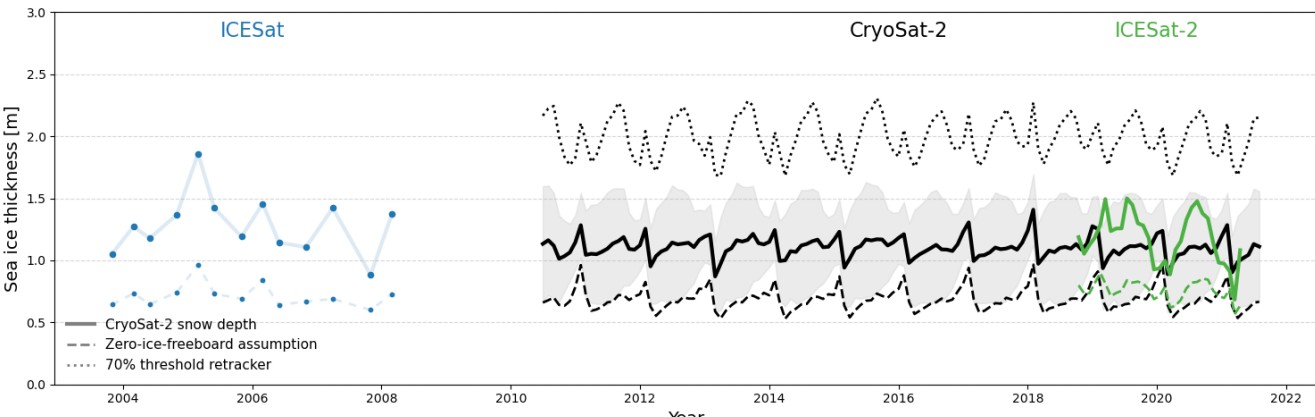

**Figure 10.** Combined pan-Antarctic sea ice thickness time series from ICESat (Kurtz and Markus 2012, blue), CryoSat-2 (CS2WFA, black), and ICESat-2 (ATL10, green). Solid lines are computed from CryoSat-2 snow depths, while dashed lines are computed using the zero ice freeboard assumption. The dotted line shows the thickness computed using a 70 % threshold retracking procedure. The shaded region gives the estimated uncertainty in CS2WFA thicknesses. Monthly CryoSat-2 snow depths were used to compute CryoSat-2 and ICESat-2 thicknesses, while an average snow depth climatology was used to compute ICESat thickness. Note that ICESat operated in discrete campaigns, and the faint blue lines shown simply connect these points and do not show the seasonal cycle.

snow freeboard over these months, which was seen in Fons et al. (2021) and thought to be related to the initialization of the CryoSat-2 waveform model. Despite the differences, it is encouraging to see general agreement in the overall mean values of

$h_{i-total}$. It is also clear that reliable, independent, and widespread validation data are crucially needed to better uncover the actual thickness distribution from Antarctic sea ice.

It must be noted that the data shown in Fig. 10 are not fully reconciled, and many important points must first be considered before trying to relate these data statistically or with more confidence. For one, the differences in geometric sampling could have substantial impacts on the freeboard - and therefore the thickness - distributions. This effect was discussed as a poten-

tial reason for CryoSat-2-ICESat-2 differences in Fons et al. (2021), but would be further complicated by adding in another sensor (ICESat) with a different footprint size. Also, the frequency differences between radar and laser must be taken into account when assessing the data, as the varying responses from sea ice and heterogeneous surfaces can impact the surface type classification, which can bias the freeboard retrievals (Tilling et al., 2019) and influence the thickness.

A final difficulty relates to validating measurements and comparing them to each other and to independent datasets. No tem-

poral overlap occurred between ICESat and CryoSat-2, which increases the difficulty in trying to reconcile the measurements. Instead, independent measurements must be used that existed for both platforms. While these independent measurements spanning ICESat and CryoSat-2 operation exist in the Arctic (e.g. the WHOI Beaufort Gyre moorings, WHOI 2018), they do not currently exist for Antarctic sea ice. At the time of this writing, CryoSat-2 and ICESat-2 are in coincident operation with the



CryoSat-2 orbit optimized for more frequent overlaps with ICESat-2 in the Southern Hemisphere. These CRYO2ICE data
(European Space Agency, 2018) will be especially useful in reconciling and extending the sea ice thickness record.

## 6   Conclusions

In this work, estimates of Antarctic snow depth on sea ice and sea ice thickness derived from CryoSat-2 have been shown.
The physical model and waveform-fitting process introduced in Fons and Kurtz (2019) and improved in Fons et al. (2021) was
applied to all CryoSat-2 data over the Southern Ocean between July 2010 and August 2021, and results were aggregated into
monthly and seasonal-averages.

Pan-Antarctic monthly averaged snow depths for the year 2019 derived from this method showed a high bias when compared
to that from KK20, ranging between 0.2 to 7.5 cm. Seasonal means from the entire CryoSat-2 period range from 16 in autumn
to 25 cm in summer.

The retrieved freeboard and snow depth data were then combined to estimate sea ice thickness. Spatial patterns appeared as
expected (similar to that from KK20, Kurtz and Markus 2012, and others), and had climatological monthly mean values over
the CryoSat-2 period ranging between 0.95 and 1.26 m. These values are potentially on the thick side, especially compared to
estimates from Worby et al. (2008), as one would expect thinner ice during certain months of new ice growth. However, the
retrieved thicknesses tend to agree with other satellite-based estimates, specifically those from Xu et al. (2021) and KK20. A
time series from 2010 to 2021 was constructed that shows some interannual variability, but highlights a consistent seasonal
cycle and range of values.

Trends in sea ice thickness and volume were uncovered by applying the Theil-Sen estimator to regional and seasonal average
values across the 11-year time series. Statistically significant trends included positive trends in the Pan-Antarctic summer and
Am-Bel winter and annual-average (0.4, 2.3, and 0.9 cm yr$^{-1}$, respectively) and negative trends in the Ross Sea autumn, the
Eastern Weddell winter, and the Western Weddell autumn and annual-average (-0.3, -0.8, -2.6, and -1.6 cm yr$^{-1}$, respectively).
Despite small pan-Antarctic trends over this time period, regional trends and pan-Antarctic trends since 2014 show larger
magnitudes.

Overall, this work has shown a decade-plus-long time series of Antarctic sea ice thickness from CryoSat-2 using snow
depths and snow freeboards retrieved from the CS2WFA method, complimenting the snow-freeboard-derived Antarctic sea ice
thickness record that began with ICESat (Kurtz and Markus, 2012) and continues with ICESat-2 (Kacimi and Kwok, 2020; Xu
et al., 2021). It is clear that more independent validation measurements are required to better get a sense of the "true value" of
Antarctic sea ice thickness, which would help in combining estimates from these three instruments into a cohesive time series.
Additionally, future work is necessary to establish biases between the instruments and reconcile thickness estimates in light of
the geometric sampling and frequency-related discrepancies. The current temporal overlap between CryoSat-2 and ICESat-2
observations is invaluable for comparison and validation, and is even more beneficial now that the CRYO2ICE campaign has
been optimized for the Southern Hemisphere (European Space Agency, 2018). Reconciling these observations will lead to a
better understanding of recent changes in Antarctic sea ice thickness.



*Data availability.* Monthly average grids of freeboard, snow depth, and thickness estimates from CS2WFA between July 2010 and August 2021 are provided as netCDF files on Zenodo: https://doi.org/10.5281/zenodo.7327711. ICESat thickness data are from Kurtz and Markus (2012) and can be found at https://earth.gsfc.nasa.gov/cryo/data/antarctic-sea-ice-thickness. ICESat-2 thickness data are estimated from the
V5 ATL10 product (https://doi.org/10.5067/ATLAS/ATL10.005). Comparison thickness estimates come from Kacimi and Kwok (2020), Worby et al. (2008), and Xu et al. (2021).

*Author contributions.* SF devised the study and carried out the analysis. NK developed the original waveform model, while updates to the model were made by SF, NK, and MB. SF wrote the text with contributions from NK and MB.

*Competing interests.* The authors declare no competing interests.

*Acknowledgements.* The authors would like to thank the CryoSat-2 science team for delivering quality altimetry data, as well as James Carton at the University of Maryland for the productive conversations regarding this work.



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
