# Peer review of "A decade-plus of Antarctic sea ice thickness and volume estimates from CryoSat-2 using a physical model and waveform-fitting"

_EGUsphere, 2022_

## Referee Comment (RC2)

This paper presents an important work to not only fill in the gaps of the ICESat/ICESat-2 observations on the Antarctica sea ice thickness, but also provide a continuous ice thickness time series showing obvious seasonal cycle characteristics with CryoSat-2 from 2010 to 2021. The authors utilize a physical model and a waveform fitting method that they developed in their previous work to get snow depth and total freeboard, then the sea ice thickness and volume. This work provides a sea ice thickness dataset that could be merged with that derived with ICESat/ICESat-2 to produce a longer-term observations of circum-Antarctica sea ice, which would greatly promote global climate change studies. However, there are some concerns that need to be clarified by the authors before publication.

**Major:**

It is not clear at which step what parameters are estimated. Generally, there are several parameters involved here: total freeboard (air-snow interface elevation-derived), ice freeboard (snow-ice freeboard elevation-derived), snow depth, and ice thickness. L181 states the ice freeboard and snow depth are output parameters produced directly by CS4WFA? However, L292-293 shows the snow depth are derived from subtracting snow-ice freeboard elevation from air-snow interface elevation. So, my question is what is the input and output between each step in this study? At which step comparing with what reference dataset? I suggest supplementing a flow chart to make it clearer.

In Section 4, authors compare snow depth and thickness with other datasets. why not snow freeboard included in comparison?

L179. About the statement 'Fit parameters are discarded if the result is a "poor fit"', how much data have been discarded and what kind of data are discarded?
The similar question for L213, for what types of waveform, there is no good fit? This is important because it may better inspect the proposed method.

**Minor:**

L1." a physical waveform model and a waveform-fitting method to estimate the snow depth and snow freeboard" is misleading. What is the relationship between the physical waveform model and the waveform-fitting method and are they used for snow depth and snow freeboard respectively or as a whole? Are they the comprehensive method called CS4WFA?
The sentence is ambiguous.

L3. Is the thickness in "snow depth and thickness" snow or sea ice thickness?

L9-10. Some findings are expected after the sentence. For example, "…, showing the interannual differences between the two kinds of satellites". Or add "Results show that" before "Reconciling…"

L80-82. There may be misstatements on those literatures:
    The approach "Worby" in Kern 2016 is a static ratio between sea ice thickness and snow depth, which are seasonal empirical values from ASPeCt, which are ship-based observations.
    Li 2018 is a dynamic ratio between snow depth and sea ice thickness, which are initial guess from empirical equations between the total freeboard and snow/ice thickness by Ozsoy-Cicek Burcu, et al; Sea ice thickness retrieval algorithms based on in situ surface elevation and thickness values for application to

altimetry;*Journal of Geophysical Research: Oceans;*2013, 118 (8):3807-3822.

Xu 2022 is an improved version of Li 2018 with a similar strategy.

Wang 2022 uses the same 'Worby' method, i.e., the static ratio, in Kern 2016 for ICESat, while uses snow depth from AMSR-E/AMSR-2 for Envisat-based sea ice thickness retrieval.

So, I suggest the sentences to be rewritten. For example:

"With some empirical parameters, sea ice thickness can be estimated with ICESat/ICESat-2 alone. Regarding sea ice as a single ice/snow layer, the Worby method (Kern et al., 2016) uses a static ice-snow ratio which are seasonal empirical values from the ASPeCt program (Worby et al., 2008). The one-layer method (OLM, Li et al., 2018) and its improved model (OLMi, Xu et al., 2021) are proposed with a dynamic ice-snow ratio for each footprint measurement based on an initial guess from the empirical relationship between snow depth/ice thickness and snow freeboard (Burcu, et al., 2013)."

P18. Figure 7. What caused the uncertainty difference between different sea sectors?
How is the pan-Antarctic uncertainty computed?

L156. Add $\Psi$ after "a physically-modeled waveform" for Eq.2

L158. What is $h_{sd}$ in eq.4? If it is snow depth, should it be $h_s$ according to Eq.1 and Table 1?

L187 $R_n$ should be $R_0$ here?

Title of Figure 2. For "the daily linear interpolation", to avoid misunderstanding that the dashed lines are derived with daily data, it may be better written as "the linearly interpolated dashed lines between the midpoint dates are used as daily density estimates for the thickness calculation."

L262, $h_{fs}$ should be $h_{fs}$. Is there system uncertainty for snow freeboard in Eq.7?

L290. Section 4.4 missed in this sentence.

P15. Figure 4. Are the vertical dashed lines mean values? Can the mean, std, and model values in each season/month be put on the map? The same with Figure 5.
The position of figure 4 is better to be close to the text in Section 4.1. The position of Figure 8 can also be adjusted.

L366. The exception is not only Ross Sea. Am-Bel is 4 cm thicker in spring than summer.

L384. The last word "that" is 'than'?
L395. "thinner that" to "thinner than"

P21. Figure 9. Use different colors to differentiate decreasing and increasing regions instead of the uniform grey?

L476. About the discussion of the radar-laser difference in this section, another possible reason may be the high spatial resolution of ICESat/ICESat-2 than the grid/segment-averaged coarse resolution CryoSat-2. The

lower resolution may smooth higher or lower signals values.

P25. The legend in Figure 10 is "Sea ice thickness derived with CryoSat-2 snow depth" instead of "CryoSat-2 snow depth"?

L701-703. The citation of Meredith 2019 can be improved according to their suggestions: https://www.cambridge.org/core/books/ocean-and-cryosphere-in-a-changing-climate/polar-regions/8D76B8865B796C16991F7A9FB6271C2D

---

## Author Comment (AC1)

Response to Anonymous Referee #1

Summary:

This paper exploits an innovative method to use CryoSat-2 radar altimeter observations to retrieve first freeboard and snow thickness on sea ice before using both to estimate the sea ice thickness and volume - including a credible estimation of the uncertainty. The method has been developed and published in different publications and is therefore not described in-depth in this contribution. Here the focus lies in the illustration and discussion of pan-Antarctic and regional distributions of sea ice thickness and sea ice volume as based on 11 years of CryoSat-2 data - including a trend analysis.
Certainly this is an interesting and also important piece of work which broadens our knowledge about the thickness and volume distribution of Antarctic sea ice.

Dear Reviewer,

Thank you so much for your thorough comments on this manuscript. We appreciate the time and detail that you put into helping to improve this study.
Below, you will find our responses (blue) to your comments (black). We agree with many of your points, and have outlined ways in which we will improve the revised manuscript.

Thanks again,
Steven Fons, Nathan Kurtz, and Marco Bagnardi

General Comments:

GC1: You dedicate quite some part of your paper to a trend analysis. The relevance of this trend analysis is not sufficiently well motivated and not put into a credible context with the overall variability of both the Antarctic sea ice cover and its influencing factors. The added value of this analysis is not fully convincing. See my specific comments in this regard.
Thanks for this point. We agree that a good amount of this paper was dedicated to trend analysis. Our thoughts were that in presenting a long time series of CryoSat-2 estimates, we can help to provide some information on trends from a single satellite, unlike what has been done before using multiple satellites. This avoids the inter-mission biases and the associated uncertainty in previously-published trends. Additionally, we tried to avoid making claims as to how these trends would continue into the future, but simply presented the trends in the data that we have.
With that said, we do understand that stressing the trends as we did with only 11 years of data and not fully acknowledging the multi-decadal oscillations can be problematic. Therefore, we plan to tone down the discussion of trends in the revised manuscript. Instead, we will present a shortened section on e.g. "intra-decadal changes in the ice cover" and show how the ice pack has changed in the last 11 years, which is all the data we have. We can de-emphasize the

discussion on the significance of the trends. Additionally, we will be sure to mention the caveats and problems associated with considering these changes as trends that would continue in the future. In its place, more of the discussion section will be devoted to fully discussing the comparison datasets and the limitations therein (see GC2).

GC2: Your paper contains only few elements of inter-comparing your product(s) with other, independent results. Here I feel your paper has substantial potential for improvement. On the one hand, the discussion included so far in the paper based on the comparisons carried out would strongly benefit from a more critical view of i) the limitations of the intercomparison data sets used and ii) a more careful investigation and discrimination of level versus deformed sea ice and/or mean versus modal sea ice thickness values. On the other hand, key intercomparison data sets are left out, kind of limiting the credibility of the results presented - especially when keeping in mind that the authors' estimation of freeboard, snow thickness and sea ice thickness are not independent and therefore require an even more careful evaluation. See my specific comments for more information.

We definitely understand this point, but also feel that there is a general lack of reliable datasets to which these data can be compared against. The Operation IceBridge (OIB) data products, for example, do not include any snow depth or thickness estimates from CryoSat-2 underflights in the Southern Ocean. However, we do understand the importance of including more comparisons, and therefore plan to include a snow freeboard comparison with OIB (which **is** included for one flight in 2010, see specific comment below).
With regards to your point on mean vs. model values, we completely agree and will be sure to make this more clear in the revised version. This work focuses on the mean thickness distribution, and so we will be sure to mention that our comparison datasets also compute level+deformed ice thickness in their estimates. Additionally, we will include a comparison to the extended ASPeCt data, with the level ice thickness converted to 'total ice thickness' following the method in Worby et al. (2008).
As mentioned in GC1, we plan to use more of the discussion section to discuss the comparisons with independent datasets and their limitations.

Specific Comments:

Abstract:To my opinion, the abstract should contain a bit more information about the method, the product and its evaluation and less detailed information about the trend analysis results - simply because this is a short time series in a highly variable environment, possibly requiring 30+ years to derive any reliable trend information. See also GC1.
This is a fair point, however, we are simply presenting the trends in the data we have, and aren't making any claims as to how these trends would continue into the future. That said, we plan to rework (tone down) the section on trend-analysis (GC1), and will update the abstract accordingly in the revised manuscript.

L18: All three references given relate to the Arctic. I have i) difficulties to understand your choice to not directly focus on Antarctic conditions - as this is the focus of your paper and it does not read well to introduce / motivate an Antarctic focus paper with exclusively Arctic focus referenes - and ii) even for the Arctic the selection of the references given seems rather arbitrary, missing out several of the more recent literature that is available. I recommend to revise the references.

Thanks for the comment. This section is intended to introduce the importance of sea ice thickness as a whole (for things like maritime navigation and indigenous communities, which has historically been more pertinent in the Arctic), but I do see the point about focusing on the Antarctic right away. We will remove and modify the original references, and (since "climate studies of sea ice" could necessitate a long list of references) we have decided to reference the recent (2019) IPCC Special Report on the Ocean and Cryosphere in a Changing Climate, which discusses Antarctic sea ice thickness at length. This sentence will be modified to:

*Knowledge of sea ice thickness has long been important in the polar regions – from early Antarctic explorers navigating in icy waters (Herdman 1959) to indigenous Arctic communities traveling and hunting on the frozen sea (Nichols et al. 2004) – and continues to be a focus today for maritime navigation and climate studies (Meredith et al. 2019).*

Figure 1: I know, this is just a schematic figure. However, it wrongly implies that the part of the sea ice underneath the water surface is as thick as the part of the sea ice above the water surface. In addition, the thickness of the snow load almost certainly would lead to flooding of the ice-snow interface. Therefore, for the sake of displaying a more realistic schematic figure - that even lecturers might want to take from your paper - I recommend to replace this figure by one which has more realistic dimensions.

Thanks for the comment. Actually, it was meant to be a more realistic Antarctic schematic, where there tends to be a large snow load and flooding is common. That said, the draft was indeed unrealistically thin. This figure will be updated in the revised manuscript to the following:

[Figure]

In the caption, h_fs is not mentioned yet.

The term h_fs is mentioned in the caption: "…including snow depth (hs), ice freeboard (hf_i), snow freeboard (hf_s), ice thickness (hi)…"

L71: "... these estimates are both assumed to be biased high ..." --> There has been a Cryosphere Discussion paper around for a while (tc-2021-227 by Wang et al.); if I recall correctly they took an independent look (not from the producer's side) at the ESA sea ice thickness product. It might be worth a look.
Thanks for the reference. The Wang et al. work focuses on ICESat and Envisat, while the statement in the manuscript discusses the high bias found in the CryoSat-2 results only.

L121-125: ICESat had only several dedicated measurement periods while ICESat-2 has been operated continuously. I therefore assume the climatology maps have a different number of months as their baseline, i.e. for August or December it is possibly mostly ICESat-2 - aka data from 1 or 2 years, respectively, while for March it is data from one ICESat-2 year and five ICESat years. I recommend to include a short table detailing this difference in representativity of the climatology freeboard maps of the different months.
Not clear as well is how the different coverage of ICESat measurements over different months is taken into account in the respective monthly mean. Often these maps are bi-monthly maps derived e.g. half from February and half from March. How is this realized in your climatology? Did you use a Feb/Mar ICESat map for both February and March?
Good suggestion about the table. This is something we can add to the revised manuscript, which may make most sense in supplement.
While you are correct that ICESat operated in discrete campaigns (and that the campaign files cross months), the full-resolution ICESat data still contain time stamps, and therefore only data collected in a given month is included in the initialization map for that month.

L131: "area of the grid cell" --> please provide the information where you obtained the grid cell areas from. Since this data is on a polar-stereographic projection the grid cell area varies with latitude and you possibly downloaded and used the respective file from NSIDC (?)
Thanks for bringing this up, and that is correct. Grid cell areas were computed with the NSIDC files (NSIDC-0771), which was not adequately described in the manuscript. We will add this in the revised version.

L137-139: Undoubtly the Worby et al. (2008) data set is a benchmark in this direction. I note, however, that it terminates in March 2005. Have you considered to take a look at the extension of this data set available here:
https://www.cen.uni-hamburg.de/en/icdc/data/cryosphere/seaiceparameter-shipobs.html ?
Thanks for the advice - we have looked at this extension dataset, and will include a figure and some discussion comparing it to these CryoSat-2 data. We plan to adjust these level-ice estimates to be 'level+deformed' following the method outlined in Worby et al. (2008).

Table 1: In the text you state "angular backscatter efficiency"; I suggest to use the same expression in the table.

I note that snow depth seems to be given in cm while the roughness has a different unit. You could consider harmonizing this.

I am a bit puzzled about the bounds. For sigma you state bounds 0-1m; I assume this means that sigma is allowed to range between 0 and 1 m. However, for snow depth you specify a plus/minus range around the values suggested by the climatology rather than a range such as specified for sigma - otherwise the snow depth would need to range between -30 cm and +30 cm. Even when it is the range around the climatology values (which I assume) I am wondering what happens at a snow depth of 5 cm.

What is "std"?

I note that the static parameters don't have bounds even though you apply the retrieval year-round and backscatter / extinction characteristics of snow and ice may change throughout the year. Would it therefore make sense to introduce bounds here as well?

Thanks for the comments. Our responses:

- Thanks, we will update to "angular backscatter efficiency" in the revised version.
- The units will be harmonized in the revised version as well.
- For sigma, it is always initialized to the same value, and we provide bounds that are expected over sea ice (0-1m). The snow depth input varies, and therefore we can't assign a set range. Instead, (as you correctly guessed) plus/minus around the input value is used. However, we did neglect to mention that if the input is below 0.3 m, then the lower bound is 0 (i.e. does not go negative). We will update this info in the revised version.
- Std = standard deviation (to differentiate from sigma already used). We will add this definition into the caption.
- These parameters are fully static, and therefore do not change. So it wouldn't make sense to introduce bounds. While we would ideally have these as dynamic parameters and vary them by season/region with bounds, there is not enough information to accurately constrain them, or know how they vary throughout the year. We explored this further in (Fons et al. 2021, https://doi.org/10.1029/2021EA001728), but will add some info here in the revised manuscript as well.

L175: "or until ... is reached" --> What are the output parameters taken in this case? The very last one? Also, I asssume that finding a minimum residual results in a quantitatively better fit. How often does the retrieval needs to reach 100 evaluations compared to finding an adequate minimum?

Good catch, and apologies for not including this. In the event there is no convergence, the waveform is discarded. This happens very infrequently (< 0.5%). More often, a minimum solution is reached, but is a poor fit (with a large residual). The waveform will then be discarded by the filtering process. We will add this information in the revised version, as well as info on the total waveform discard percentages (per the other reviewer).

L183: "the nominal tracking bins" ... what is their difference and hence an approximation of the vertical resolution of the approach?

I'm not sure we fully understand the comment, but the nominal tracking bin is a value provided in the CryoSat-2 data products. This is the location (in range bins) to which the range from the

satellite to the surface is computed, and from which the retracking calculation is obtained. It is given as the center range bin in the CryoSat-2 handbook.

L207: "if at least three lead-type points exists within" --> I am sorry for asking this question but can successive points overlap or are they truly independent, i.e. adjacent footprints do not overlap because the along-track distance between their centers is larger than the along-track dimension of the footprint?

No problem, thanks for bringing it up. While each CryoSat-2 shot does overlap in the along-track direction (due to the large footprint), the SAR processing of the waveforms keeps the measurements fairly "independent" in the along-track direction by slicing the measurements up into strips that are fairly narrow along-track. Due to this processing, we can assume successive points are independent enough for the purposes of finding candidates SSH points.

L237: I could have asked this question earlier in the context of the SSH approximation: For that approximation you need a minimum of 3 valid points within a 10-km segment. And then you first compute the parameters mentioned along track, i.e. for each valid floe-type point along-track, and then perform the gridding?! For the latter, does that have to be a minimum of valid floe-type points from which the parameters mentioned are computed? I can imagine that there are seasons and regions where you may have quite a number of valid lead-type points and a lot of mixed-type points but only few floe-type points.
Finally, the SSH derived is representative for 10-km segments, i.e. in the worst case a step-function in SSH, or is this derived using a running 10-km segment possibly providing a smoother representation of the SSH?

This is a good point to bring up. The parameters are computed for each waveform, so there is no minimum amount in that case. For the gridding, we neglected to mention this (sorry), but we use 5 waveforms as a minimum number of waveforms to create a grid (following Kurtz et al. 2013). We will add this in the revised version.
Also, the 10-km SSH is done using a running segment, since you are correct that discrete 10km segments would create a step function SSH. We will add this clarification to the revised version as well.

L278/279: While following the approach of Spreen et al. (2009) is at first place good, I am wondering whether it would make sense to back this value up by looking into sea-ice concentration uncertainty information that is provided with the OSI-450/OSI-430-b CDR/iCDR sea ice concentration data set (which includes smearing uncertainty contributions) or with the NOAA/NSIDC SIC CDR (even though this is basically a modified standard deviation)? Another source you could look into in this regard is this one:
https://egusphere.copernicus.org/preprints/2022/egusphere-2022-1189/
Thanks for the comments. While we acknowledge that these methods would improve the sea ice concentration uncertainty estimate, we stress in the paper that this is a simple approach used to provide some idea of the volume uncertainty, and feel that the conservative estimate of 5% works for this purpose. Future work that is focused on more fully constraining the uncertainties in sea ice thickness/volume retrievals would benefit from these studies.

L310: "more validation data" --> I would call your comparison to the KK20 product an inter-comparison. It is not a validation as the KK20 data is just one possible result of combining two satellite data sets to obtain a snow thickness product. For an evaluation or validation one would need ground-truth data which the KK20 data certainly is not. Hence, my suggestion is to stress that we need "validation data" (hence delete "more" as this implies that the KK20 data are already validation data) of the type ground-based measurements to really be in the position to perform a validation.

Very good point, thanks. We will make this change in the revised version.

Figure 3: I am wondering whether it would make sense to provide an estimate of the actual number of values per month as I would assume that the number grid cells contributing to a February value in the Indian Ocean sector differs considerably from the region W-Weddell.
I note that the name for sector "Amundsen-Bellingshausen Sea" has different flavors. Consider using one.

Yes, it probably makes sense to include this information. However, this figure is mostly a comparison to KK20, and therefore both sensors would experience similar relative amounts of measurements in each season. We will add in some text acknowledging that comparing regions must take into account the different number of values per month.
Good catch with the "Amundsen-Bellingshausen". We will ensure the revised manuscript is consistent, sticking with "Amundsen-Bellingshausen Seas" as the full name, abbreviated to "Am-Bel" in most of the text.

L334-338: "In Fig. 4 ... in Fig. 3)." --> How do these results compare to the snow thickness retrievals based on ICESat data in Kern and Ozsoy-Cicek (2016)? Didn't they also show an increase in the average snow thickness from autumn to winter to spring - in contrast to the snow thickness values derived using a modified version of the Markus and Cavalieri (1998, online 2013) approach?

I believe Kern and Ozsoy-Cicek (2016) looked at changes from Winter to Spring (and comparisons showing thicker spring snow depths than AMSR-E), but you're right that this would be useful here. We will add in this reference and our broad-scale comparisons to that work in the revised version.

Figure 4: The dashed line in the inset histogram denotes what?
The legend of the maps as well as the histograms is in meters. I suggest to then also set the binsize to 0.02 m.

The dashed line denotes the histogram mean (text to be added in the revised version). And, good point about the units. We feel 2cm is easier to understand that 0.02m in a figure caption, but also get the need for consistency. We will update this in the revised version.

L355 / Figure 6: I am wondering about the real information content of these probability distribution values given the fact that the number of observations per region / season varies so much. Did you consider normalizing the histograms to 1?

We did, yes. The histograms are currently normalized so that the area under the histogram integrates to one, which is generally found to be a robust way of normalizing the data.

Figure 5: Same comments as I had for Figure 4.

In addition: please swap "Spring" and "Autumn" below the bottommost row of panels.

Good catch! Thank you. These will be swapped in the revised version, and the dashed-line explanation will be added to the figure caption.

L363-368: "Despite ..." --> I have repeatedly used the Worby et al. (2008) data set (and its extension mentioned further up in my comments) for inter-comparison purposes and am well aware of its value. I am wondering, however, whether some additional information needs to be given here to underline how vague that information can be. These data have an observational negative bias because ships tend to avoid thicker sea ice. In summer, floes break different under the action of the ship's hull reducing integrity of level ice made of rafted ice floes. In summer, pancake ice which makes a substantial fraction of the observed sea ice cover, is essentially lacking. Leads, often followed by the ships, are covered by thinner ice types in summer than the freezing season. Also, in addition to these observational biases there could be biases by the regions traversed during the different seasons. For instance in the Ross sector, cruises hardly reached to the thicker sea ice parts in the eastern Ross Sea during winter and spring, simply because these areas are not accessible, but rather crossed the thinner sea ice in the sea ice export area of the Ross Ice Shelf polynya. Therefore, especially the high sea ice thickness reported in Worby et al. (2008) for summer could very well be caused by preferably entering areas with thicker sea ice compared to winter and spring. See GC2.

Thanks for this additional information, and you're exactly right that there is some additional info that could be added to this manuscript. The thought was to provide this comparison and stop short of any robust intercomparison, mainly due to these caveats in the ASPECT observations. However, we do see the utility in adding in this information. Our plan is to substantially rework the discussion section (see GC1/2), and also include more information about the comparison datasets and the limitations of each of them.

Figure 6: Are the values from Worby et al. those of the level ice or do these include the estimated contribution from ridged sea ice?

These are the average ice thickness values, so they include contributions due to ridging. They are compared to the mean of the CryoSat-2 data, which includes the ridged ice (as opposed to the modal thickness values).

Figure 7: While I was trying to understand why I have the impression that the individual mean sea ice thickness values do not add up to the pan-Antarctic mean sea ice thickness value I figured out that the scales are not the same. How important is it (for your message) to show the pan-Antarctic sea ice thickness occupying more vertical space in the figure than the individual sea ice thickness time series? Would it make sense to try to show the time series for each region with the same vertical scale?

Yes, apologies for this. We had originally made the plot with identical axes (and gone back and forth about which to include), however, you lose a lot of detail in the individual region plots, as the range can be large on some and small on the others. In the revised manuscript, we will

reduce the size of the pan-Antarctic subplot and make the scales the same, to make it easier to interpret this figure.

L390/391: "while Maksym ... thickness" --> This sounds like they used satellite microwave radiometry to estimate sea ice thickness but what they did is first of all not that simple and secondly their main statement refers to level, undeformed sea ice which is not 1-to-1 comparable to your work. I therefore invite you to check the reference one more time and to rephrase your sentence accordingly. It is important to check out which part of the sea-ice thickness distribution the respective publications refer to to be able to make appropriate statements here. In this context it might be a good idea to, in addition, introduce a discussion of modal sea ice thickness values representative of the level sea ice.

Fair point, thanks for bringing it up. We did lump these all together, when we should have added in more information on the mean vs. modal thickness from each study. We will be sure to make this distinction in the revised manuscript, and, seeing as we are primarily focused on the mean thickness (level + ridges) for the rest of the work, we will ensure that any dataset that is being compared is also concerned with the mean thickness.

L398/399: "Williams et al. ..." --> You might find it enlightening to again take a look into Kern et al. (2016). Even though their 1-layer method results are possibly biased and rather refer to the total (sea ice plus snow) than the "true" sea ice thickness, the intercomparison of the other methods (including the ZIF) seems to provide a possible range (at least for the ICESat measurement period) of sea-ice thickness values obtained using different methods.

Good point – Kern et al. does show examples of estimates that are indeed thicker than other observations, and could provide some argument for a 'thicker-than-expected' ice thickness. We won't try to directly compare to these results (due to the different time periods), but can reference this work in the revised version similar to what was done with Kurtz and Markus (2012).

Section 4.4: In light of the substantially larger (and known) variation of the Antarctic sea ice cover - compared to the Arctic - I have a conceptual problem with dedicating a full sub-section to a trend analysis of an eleven years long time series. This looks like somebody wants to investigate an eleven years long precipitation time series of the U.K. in light of trends. But it is of course your decision to keep or delete this part of the manuscript. In case you keep it I strongly recommend to - beyond statistical significance estimates - state clearly that any trend found for these eleven years can simply be the part of a multi-decadal variation that cannot be resolved yet with the existing record of CrysoSat-2 sea ice thickness and volume observations. This would be a good motivation to i) discuss your results even more in the context of the work of other studies; ii) to advertize more work needs to be done to include Envisat and ERS1/2 RA altimeter data analysis to extent the time-series; iii) to advertize your own sub-section about expanding the CS-2 time series back in time to the ICESat periods.

This is a very valid point, and will be reworked substantially in the revised version of the manuscript. See GC1 for a full response regarding this trend analysis.

L420: "contains at least four years of data" --> Is there any constraint as to when these four years need to contain data? Is it possible that all data are from the first 4 years?
There is no constraint applied, other than having a minimum amount of data. This really only impacts the ice margin, where the extent changes more drastically year-to-year, and does not impact the "central" ice pack nearly as much. This section will be re-worked in the revised version, but can also include a disclaimer that the 4 years could come from any time period.

L432: "Holland (2014)" --> How many years of your 11-year period overlap with the data used by Holland (2014)? Are those results therefore compatible with your results?
There is not much overlap between the two datasets – just part of the year 2010. This is why there is not much discussion given to this comparison (also because of the differences inherent in a modeling study vs. observations). However, we will make a more clear note of this in the revised version.

L432/434: Both, Garnier et al., (2022) and Xu et al. (2021) used data from a longer time series, didn't they? What is then the added value of performing such a trend analysis over a shorter time period? This is not entirely clear to me.
It was mainly done to showcase the trends one can retrieve from a single sensor. All other studies either combined data from multiple sensors (the ones you listed), or showed a time series from a single sensor that was shorter than ours. A more detailed answer and plans for the revised manuscript can be found in GC1.

L441-443: How many years of CryoSat-2 data did Kwok and Cunningham (2015) use in their analysis? I checked it out: It is four winters. You investigate 11 years. I don't think your current writing (and citing that paper) does support further discussing the impact of an analysis of 11-years worth of sea ice thickness and volume in the Southern Ocean.
We're not sure we fully follow your argument. They do include 4 years of CryoSat-2 data, but calculate trends from ICESat2 to CryoSat-2 (2003- 2013). This citation was included simply to provide an estimate of sea ice thickness trends in the Arctic, as a way to compare to what we're finding in the Antarctic. Nevertheless, our trend discussion will be reworked in the revised manuscript, see GC1.

Below in this sub-section you will find more comments going into this direction. All I wish to trigger with these is to encourage you to one more time critically think whether the message you provide here is compelling, sustainable and worth the effort. Does it send out the right signal in view of already existing work and in view of what we know about the length time series of geophysical parameters should have in order to provide a meaningful statement about climatological features such as trends? See GC1.
Your point has been received. See GC1 for a full response about the trend analysis and plans for the revised manuscript.

L455-457: "However, the same ... since 2014" --> Certainly. And if you shorten the time period even further, e.g. to a 4-years like Kwok and Cunningham did, then you will find an even larger

decrease in sea ice thickness or volume for 2014-2017 while you may find an increase in sea ice thickness or volume for 2011-2014 and 2017-2020. Fine. And?

We are just presenting trends in the data we have. We aren't saying that these larger trends will continue or are representative of a long time period. Just that this is what the trend is over a given time range. See GC1 for a complete response.

I find it kind of dangerous to refine the temporal granularity of such trend analysis in an area such as the Southern Ocean being influenced by at least three multi-decadal oscillations plus El Nino/La Nina events. I agree, Kwok and Cunningham (2015) did it with an even shorther time series, Kurtz and Markus (2012) as well ... but what did we learn from these?

I think in this case we (and the above cases) are presenting the trends in the data that are available. We're not trying to extrapolate past or future trends from these data. See GC1 for a complete response.

L463/464: "modeled studies into ... scenarios" --> Certainly. But this is not a surprizing finding and, in addition, it requires first some more work still to be done improving those models - see Roach et al., 2020, Geophys. Res. Lett., 47, who for good reason first looked at the Antarctic sea ice area in CMIP6 models finding it not well represented.

Agreed. This section will be re-worked, but if it remains we can add a statement that these models need to be improved as well.

L472/473: "A longer-term time series ... implications." --> Exactly. Two other studies exist (almost certainly there are more in the meantime) that already looked into longer time series which complicates to see the immediate added value of your investigation in comparison to their studies.

See GC1.

L497-501: "Likely ... estimate Antarctic sea ice thickness." --> I have two comments here. The first one is related to whether you also looked into the work of Ricker et al., 2015, Impact of snow accumulation on CryoSat-2 range retrievals over Arctic sea ice: An observational approach with buoy data, Geophys. Res. Lett., 42. While being for Arctic conditions that work might be further enlightening with respect to your observations.

Yes, we are familiar with this work, which could perhaps inform what we're seeing here. However, we feel that there is plenty of evidence for overestimation of sea ice thickness in the Antarctic using the threshold retracking method (works cited in text) and that tying in more Arctic work would overcomplicate things in this section.

The second comment is about the observation that in the time series of h_i_70 the primary maximum mean sea ice thickness is not occurring in February anymore but occurs in late winter / spring in all but one year. What does this tell us in light of the fact that the primary maximum now occurs close to the maximum sea ice coverage - involving a large fraction of seasonal sea ice with different surface properties than encountered in February?

This is a very good point, and one that was not discussed in the original manuscript version. It could be a number of reasons, one of which ties in to your above point of snow accumulation.

Since this method is doesn't explicitly account for the impacts of increasing snow load, it could retrieve an increasingly anomalous thickness as snow accumulates throughout the season. We can add in some discussion on this differing yearly maximum in the revised version.

L505/506: "could come from ... its lifetime" --> I am aware of these changes but at the same time I am wondering i) which release of the ICESat GLAS data you used for your re-processing of the ZIF sea ice thickness values and ii) whether you did not correct for the different gain values that are reported along with the ICESat data?
These data are taken from https://earth.gsfc.nasa.gov/cryo/data/antarctic-sea-ice-thickness, using the method described in Kurtz and Markus (2012), which have not been reprocessed since publication.
These data are not corrected for different gain values, however, they also only cover the years in which the data were more reliable (i.e. there are no data from later years when the gain values were much higher.)

L525 / Section 5:

I absolutely agree with you that it would be really nice to have ground-based observations that cover all three sensors' observation period. But we know that this is not possible. The only data sets I am aware of that covers all three sensors contain only estimates of the sea ice thickness: the ASPEcT data set and its extension mentioned further up. Arctic studies often tend to look into PIOMAS to see whether there is long-term consistency in the estimates. I am not deep enough involved into such studies to know whether GIOMAS data would be a viable alternative for the Southern Ocean.
However, apart from these considerations, I am missing a more thoughtful evaluation of your sea ice thickness data / product for the CryoSat-2 period used. I have several concerns. One is the apparent lack of adequately discriminating between modal (level) and mean (level + deformed) sea ice thickness values in those parts of your intercomparisons where such a discrimination would be possible (e.g. the Worby et al., 2008 data). In that context I note again that you could have used the extended version of these data noted earlier in addition - even though these do not contain this discrimination into level and level+deformed ice.
Thanks for this. We will include a comparison to the extended ASPeCt dataset in the revised version, and be sure to discuss the caveats along with using such a dataset (including but not limited to level vs level+deformed ice).

In this context I would like to remind you to adequately discuss the limitations of the data you used for your intercomparisons presented in this manuscript - as voiced further up in the context of the Worby et al. (2008) data.
Thanks, a section of discussion on these limitations will be added to the revised version, see the GCs.

What I am missing is consideration of Operation Ice Bridge data in your evaluation and discussion of the quality in this manuscript. There is a substantial amount of data available and

even though flights mostly cover the Weddell and Bellingshausen Seas these are nevertheless a very valuable source for the evaluation of your product. Other air-borne data exist, such as helicopter-borne electromagnetic sounding but I am in fact not sure how many of these would be available within the CryoSat-2 period. For sure researchers organized from New Zealand obtained data in the southern Ross Sea.

While Operation IceBridge did indeed collect a substantial amount of measurements over Antarctic sea ice, what we are missing is a snow depth/ thickness product for these data. There are only 6 flights in which snow freeboard data are provided by NSIDC, covering October 2009 and 2010 (3 during CryoSat-2 period). Of these, only one performed an underflight of CryoSat-2, though it only provides snow freeboard (no snow depth nor thickness estimates). While one could generate their own product, it would require a separate study to validate and trust the results. More OIB data are planned to be processed in the future, but this is all that is available currently.

Nevertheless, the other reviewer also suggested further comparison, and this underflight would be a good addition. In the revised version, we plan to provide a comparison to OIB snow freeboard from Oct 28 2010 in the Weddell sea.

In short, in view of recommendations I conveyed to other authors with a similar manuscript profile my main recommendation for you and your section 5 is to put more emphasis on more critically discussing the reliability of your results rather than discussing trends.

We appreciate the comments, and plan to substantially rework the discussion of trends and reliability of results in the revised version. See GC1/2 for more information.

Editoral Comments / Typos:

L25: "snow freeboard" --> You could add that here the assumption is that the dominant scattering comes from the snow surface.

Good point. This does help with the clarification and will be added in the revised manuscript

Equation 2: I recommend to add the information that the second term actually results in a reduction of the sea ice thickness computed by the first term alone - which is opposite to Equation 1 - and which particularly in the Antarctic - the focus of your paper - is important to consider as snow freeboard might equal the snow thickness or may even be smaller than that in case of flooding.

This is a very good point. Some information to this effect will be added in the revised version.

"Kurtz and Markus, 2012" and "Kwok, 2011" are references in which one can find these two equations - however, I am wondering whether it wouldn't make more sense to go back to those publications where these equations were developed / introduced first ... which might be the Laxon et al. paper from 2003 in case of Equation 1 and one of the earlier Kwok (et al.) papers for Equation 2.

Good point, and thanks for the suggestion. It appears to be Laxon et al. 2003 for equation 1 and Zwally et al. 2008 for equation 2. We will use these in the revised version.

L36: ICESat facilitated "snow freeboard" measurements. Please correct.
This "sea ice freeboard" was meant more in the general "freeboard of sea ice", but I do see the confusion. We will correct this in the text.

L39-42: "In most of these ... Kurtz et al., 2009)" --> I suggest to place the Warren et al. reference behind "1954-1991"; otherwise it reads as if Warren et al. (1999) have used that climatology to convert freeboard to thickness.
Good point, thanks. This will be modified in the revised version.

I further suggest to not highlight that Kurtz et al. (2009) used snow thickness data from passive microwave sensors (which by the way do not provide "lower resolution" snow thickness data compared to the Warren et al climatology being based on interpolation using a polynomial function anyways) - simply because this is just one of the alternatives used by the various other groups already cited. How important it is for the Antarctic focus of your paper to introduce the reader to potential alternatives to the Warren et al. climatology which is not existing in the Antarctic?
The 'lower resolution' was not meant to be a comparison to the Warren Climatology, but instead a general statement on low-resolution snow depth data. We do see how this is confusing and will modify it in the revised version.
We do feel it is important to bring up all the various snow depth data used in the Arctic, as a way to drive home the contrast between the available data in the two hemispheres. That said, it may not add much to the overall message, and can instead highlight in our revised version that snow models are used more frequently (recently) in the Arctic.

L43: ICESat-2 --> Did you overlook the contributions by Kwok (et al.) - who also combined Cryosat-2 and ICESat-2 - on purpose here?
This section is on studies retrieving freeboard/thickness from a single sensor, either CryoSat-2 or ICESat or ICESat-2. Kwok et al. 2020 uses combined ICESat-2 and CryoSat-2, which is the topic of a later paragraph, found on line 87.

L45: "have found success in estimating sea ice freeboard over Arctic sea ice" --> In light of the fact that most of the studies you cited had freeboard-to-thickness conversion as their ultimate aim, I am wondering whether you might want to rephrase this along the lines: "were succesful in retrieving sea ice thickness from sea ice freeboard estimates over Arctic sea ice" ... or the like.
Good suggestion, thanks. That does make more sense given the focus of this work. This will be changed in the revised version.

L57: "Markus and Cavalieri, 2013" --> This is the electronic version of the original book chapter from 1998, right? Has the content changed? If not, please check with EGUSphere how to cite to avoid the impression that this is a more recent work.
Apologies, and thanks for catching this. There was an error in the .bib file that showed an incorrect year. The correct year will be cited here in the revised version.

L74: "through the use of key snow depth assumptions" --> I might be wrong but it is only the Kurtz and Markus (2012) work which does this assumption. I therefore suggest to add something like "partly" or "for example" to make clear that assuming zero freeboad is ONE possible solution - with limited applicability though as one can figure out in the subsequently cited by you literature.

Thanks, we can make this change in the revised version.

L78: "Zero ice freeboard ..." --> In addition to citing Willatt et al (2010) you could also include Ozsoy-Cicek et al. 2013, JGR-Oceans.

Good suggestion – this will be added in the revised version.

L79: Regarding this underestimation you could have cited the earlier study by Kwok and Maksym from 2014 (JGR-Oceans) using OIB data; also Kern et al. (2016) performed intercomparisons between different retrieval approches, Kurtz and Markus being on of these.

Thanks for the suggestion. Citing these works here is indeed a good idea, and will be done in the revised version of the manuscript.

L95 "utilize CryoSat-2" --> It would not hurt to also mention Envisat here because with that one would have an uninterrupted time series produced using an independent sensor from 2003 through today (see also Paul et al., 2018).

Good suggestion - we will mention Envisat here in the revised manuscript, and still highlight the importance for having CryoSat-2 to fill the gap.

L127: "is based off of" --> I would have written "is based on" ... but I am not a native English speaker ...

Yes, I believe you are right. Thanks for catching it – we will make the change in text.

L160: You might want to change the font of P, I and p so that it matches "v" and equation (4).
L187: "$R_n$" needs to be "$R_0$" ?

This will changed, thanks.

L197: I am not sure I would throw the Schwegmann et al. paper into one pot with the Paul et al one because the latter used a considerably modified methodology. Hence citing Paul et al might be sufficient here.

Fair point, thanks. We will remove the Schwegmann reference here in the revised manuscript.

L239-241: You could consider to delete the information about which algorithm you used and how you compute the sea ice area because you described this earlier.

Good point, thanks. This will be removed in the revised version of the manuscript.

L247: You might want to add that the demarcation in longitude is given in the unit "degrees East".

Thanks for the suggestion – I agree, and will make that change in the revised manuscript.

L262: The "s" in h_fs needs to be put in sub-script mode.
This will be changed in the revised manuscript.

L267: "h_f" at the end of the line needs to be "h_fs"?
This will be changed, thanks.

L293: "sections" --> "section"
This will be changed, thanks.

L301: What is "IQR"?
This is the inter-quartile range, which (we realize) was defined in the figure caption but not in the text. We will add in a line to provide the definition.

L308/309: "Despite ... nevertheless" ... I guess one of these is enough; I'd discard the "nevertheless".
Fair point, thanks. We will remove "nevertheless".

L325/326: "This could cause ... anomalous snow depths" --> I am wondering whether you could narrow this down towards that the snow-ice interface will most likely be located higher in the snow pack and then also state that this will lead to anomalously low snow thickness values?
Good point, this can definitely be clarified more along the lines of what you suggest. We did mean anomalously low snow thickness values, but (obviously) just left it at anomalous. This will be updated in the revised version.

L379/380: You are refering to "basal growth" here. For Southern Ocean sea ice a substantial portion of the sea ice volume (up to 1/3 in some places) is actually made of snow ice, i.e. snow that was first flooded at the ice-snow interface and then re-froze. This is not a basal growth. One solution could be to simply write "growth".
Thanks for the suggestion – yes, we called it 'basal growth' but indeed did mean to include other types of growth processes as well. We will change this to 'growth' in the revised manuscript and make sure to reference the possibility of growth through snow-ice formation.

L384: "in ice thickness" --> in pan-Antarctic sea ice thickness"
Good suggestion – this clarification will be added in the revised version.

"more that" --> "more than"
This will be corrected.

L394: A reader would be happy to be reminded what this correction factor does and when it is applied.
Yes, agreed. We will include a note about what the correction factor is and how it was used in that study.

L395: "was estimates" --> "was estimated"

This will be corrected.

L459: Sometime you use pan-Antarctic with a capital "P" sometimes not. You might decide for one version of how to write it. I don't know actually what would be correct grammatically. Thanks for pointing this out – it seems I also was not sure what would be correct grammatically. We've confirmed that it should be written with a lowercase "p", and will make that change throughout.

L484: "about" --> "around"
This will be changed.

L510: "h_i-total" is what?
This is the thickness of ice derived from the total freeboard. We realize this was defined as h_i-sfb in equation 2, so will make this change here. Additionally, we will harmonize the use of h_i-zif and h_i-oifb in this section with equations 1-3 in the revised version.

L550/551: "It is clear that ... sea ice thickness," --> I encourage you to also include "snow thickness" here.
Very good suggestion, thanks. We will include a mention of snow thickness here as well.

L611: You might want to replace this reference by the paper published in Earth and Space Science, 8(7), 2021 to have the link to the peer-reviewed version of your work.
Thanks for catching that, it is indeed an old link that was carried through in the .bib file. We will update this link to the final, peer-reviewed version of that manuscript.

---

## Author Comment (AC2)

Response to Anonymous Referee #2

This paper presents an important work to not only fill in the gaps of the ICESat/ICESat-2 observations on the Antarctica sea ice thickness, but also provide a continuous ice thickness time series showing obvious seasonal cycle characteristics with CryoSat-2 from 2010 to 2021. The authors utilize a physical model and a waveform fitting method that they developed in their previous work to get snow depth and total freeboard, then the sea ice thickness and volume. This work provides a sea ice thickness dataset that could be merged with that derived with ICESat/ICESat-2 to produce a longer-term observations of circum-Antarctica sea ice, which would greatly promote global climate change studies. However, there are some concerns that need to be clarified by the authors before publication.

Dear Reviewer,

Thank you so much for your comments on this manuscript. We appreciate the time you put in as well as your contributions to improving this study.
Below, you will find our responses (blue) to your comments (black). We agree with many of your concerns, and have outlined ways in which we will improve the revised manuscript.

Thanks again,
Steven Fons, Nathan Kurtz, and Marco Bagnardi

Major:
It is not clear at which step what parameters are estimated. Generally, there are several parameters involved here: total freeboard (air-snow interface elevation-derived), ice freeboard (snow-ice freeboard elevationderived), snow depth, and ice thickness. L181 states the ice freeboard and snow depth are output parameters produced directly by CS4WFA? However, L292-293 shows the snow depth are derived from subtracting snow-ice freeboard elevation from air-snow interface elevation. So, my question is what is the input and output between each step in this study? At which step comparing with what reference dataset? I suggest supplementing a flow chart to make it clearer.
Thanks for the suggestion, and apologies for the confusion. Some of these processes can be described in different ways. For example, the snow depth **is** an output parameter from the model, but is identical to subtracting the interface elevations and therefore can be "calculated" either way. Additionally, the snow-ice interface tracking point is output from the model, which is used to find the snow-ice interface elevation (via equation 5) and then ice freeboard. The updated manuscript will better describe some of these processes. In addition, we will add in a flowchart similar to the following:

[Figure]

Where ovals show input/output data (green is start, red is end, white are inputs, grey is discarded), diamonds show decision/filtering processes, white squares show calculations/other processes, and blue squares show 'milestones' (inset orange squares for lead-type milestones, blue for floe-type milestones).

In Section 4, authors compare snow depth and thickness with other datasets. why not snow freeboard included in comparison?
This is a fair question. As mentioned in the text, a comprehensive snow freeboard comparison was done using ICESat-2 data in Fons et al. (2021; *Earth and Space Science*). We point readers there for this comparison, but I understand that it still feels lacking in this work. The other reviewer had similar thoughts, so we plan to add in a snow freeboard comparison to Operation IceBridge data from a flight in the Weddell Sea in October 2010 to the revised version.

L179. About the statement 'Fit parameters are discarded if the result is a "poor fit"', how much data have been discarded and what kind of data are discarded?
Good question, this is info that should have been included. The number varies from orbit-to-orbit, but on average 86% of waveforms are kept (meaning 14% are filtered out). That number is higher for floe-type waveforms (21% filtered out). Only <1% of lead-type waveforms are filtered out. These numbers tell us that waveforms that are filtered out are typically "messy", meaning that they have multiple peaks (due to scattering from off-nadir leads) and/or don't fit the typical profile of a return floe-type waveform. Lead waveforms are typically specular and without many off-nadir peaks, and therefore are less likely to be filtered.
We will add these values to the revised manuscript.

The similar question for L213, for what types of waveform, there is no good fit? This is important because it may better inspect the proposed method.

Good question, with a similar answer to above. Poor-fitting waveforms typically have multiple peaks, usually brought on by off-nadir leads ( Kurtz et al. 2013, Tilling et al. 2018).  This is a common problem with many CryoSat-2 retracking methods, as the wide footprint is susceptible to returns from far off-nadir. We do understand that filtering these waveforms out could potentially bias the retrievals, especially for sea ice close to leads. However, quantifying this bias would be very difficult. Overall, we will add in some info towards this point in the revised version.

Minor:

L1." a physical waveform model and a waveform-fitting method to estimate the snow depth and snow freeboard" is misleading. What is the relationship between the physical waveform model and the waveform-fitting method and are they used for snow depth and snow freeboard respectively or as a whole? Are they the comprehensive method called CS4WFA? The sentence is ambiguous.

You are right that they are all part of a comprehensive method that we're calling CS2WFA for simplicity. We construct a model and use the waveform-fitting approach to estimate output parameters, which includes snow depth and snow freeboard. We can modify this in the revised version to something along the lines of: "We estimate the snow depth and snow freeboard of Antarctic sea ice using a comprehensive retrieval method (hereafter referred to as CS2WFA) that consists of a physical waveform model and a waveform-fitting process to fit the modeled waveform to CryoSat-2 data. "

L3. Is the thickness in "snow depth and thickness" snow or sea ice thickness?

It is referencing the sea ice thickness, and we will add this in for more clarification.

L9-10. Some findings are expected after the sentence. For example, "…, showing the interannual differences between the two kinds of satellites". Or add "Results show that" before "Reconciling…"

Fair point. We plan to revise the abstract in the updated manuscript based on both reviewers' comments, however, if this sentence remains it will change it to something like: "We place these thickness estimates in the context of a longer-term, snow-freeboard-derived, laser-radar sea ice thickness time series that began with ICESat and continues with ICESat-2, and show interannual differences between these satellites."

L80-82. There may be misstatements on those literatures:

The approach "Worby" in Kern 2016 is a static ratio between sea ice thickness and snow depth, which are seasonal empirical values from ASPeCt, which are ship-based observations.
Li 2018 is a dynamic ratio between snow depth and sea ice thickness, which are initial guess from empirical equations between the total freeboard and snow/ice thickness by Ozsoy-Cicek Burcu, et al; Sea ice thickness retrieval algorithms based on in situ surface elevation and thickness values for application to  altimetry;Journal of Geophysical Research: Oceans;2013, 118 (8):3807-3822. Xu 2022 is an improved version of Li 2018 with a similar strategy.

Wang 2022 uses the same 'Worby' method, i.e., the static ratio, in Kern 2016 for ICESat, while uses snow depth from AMSR-E/AMSR-2 for Envisat-based sea ice thickness retrieval.

So, I suggest the sentences to be rewritten. For example:

"With some empirical parameters, sea ice thickness can be estimated with ICESat/ICESat-2 alone. Regarding sea ice as a single ice/snow layer, the Worby method (Kern et al., 2016) uses a static ice-snow ratio which are seasonal empirical values from the ASPeCt program (Worby et al., 2008). The one-layer method (OLM, Li et al., 2018) and its improved model (OLMi, Xu et al., 2021) are proposed with a dynamic ice-snow ratio for each footprint measurement based on an initial guess from the empirical relationship between snow depth/ice thickness and snow freeboard (Burcu, et al., 2013)."

Thanks for this suggestion. The idea was to categorize all of these based on their use of a snow depth/ice thickness ratio, but we do see how the way it is currently written simplifies the description of these studies. We also appreciate the suggestion, and will change this in the revised manuscript to something very similar:

"Other studies have used a ratio between the snow depth and sea ice thickness to estimate the ice thickness from laser altimetry. The Worby method (from Kern et al. 2016) used a static snow-ice ratio derived from seasonal empirical values from the ASPeCt program (Worby et al. 2008), while the one-layer method (OLM, Li et al., 2018) and its improved model (OLMi, Xu et al., 2021) use a dynamic snow-ice ratio for each footprint measurement based on an empirical relationship between snow depth/ice thickness and snow freeboard (Burcu, et al., 2013)."

P18. Figure 7. What caused the uncertainty difference between different sea sectors? How is the pan-Antarctic uncertainty computed?

The difference in uncertainty in the different sectors comes from the difference in the snow freeboard, snow depth, and snow depth uncertainty between the sectors, since these are the variables in Equation 7 that vary spatially. The pan-Antarctic uncertainty is calculated simply as an average of the uncertainty of all grid cells basin-wide. This is a simplistic way to do this, but provides some estimate of where we expect the uncertainty to be. We plan to modify this figure in the revised manuscript (to answer the other reviewer's comments), but will also include a better mention of how the pan-Antarctic uncertainty is calculated.

L156. Add $\Psi$ after "a physically-modeled waveform" for Eq.2

Good idea – this will be added.

L158. What is hsd in eq.4? If it is snow depth, should it be hs according to Eq.1 and Table 1?

That is correct, and will be changed in the revised version.

L187 Rn should be R0 here?

Yes, thanks for catching this. It should indeed be R0, and we will make that change in the revised version.

Title of Figure 2. For "the daily linear interpolation", to avoid misunderstanding that the dashed lines arederived with daily data, it may be better written as "the linearly interpolated dashed

lines between the midpoint dates are used as daily density estimates for the thickness calculation."

Thanks for the suggestion - that is a small but important distinction. We will make these changes in the caption of Figure 2.

L262, hfs should be hfs. Is there system uncertainty for snow freeboard in Eq.7?

Thanks for pointing out the formatting error – we will make that change.
Equation 7 is based on Petty et al. (2020) and Ricker et al. (2014), and is a simplistic way to provide some measure of uncertainty. It does not include an estimate of systematic uncertainty for freeboard, since it is not reliably known. While we acknowledge that an estimate of snow freeboard uncertainty would help to better constrain the total thickness uncertainty, the current equation is used until we have a better idea of the snow freeboard uncertainties. We will add in more information about the limitations of this uncertainty estimate in the revised version.

L290. Section 4.4 missed in this sentence.

Thanks – we will update this sentence to include a reference to section 4.4.

P15. Figure 4. Are the vertical dashed lines mean values? Can the mean, std, and model values in eachseason/month be put on the map? The same with Figure 5.
The position of figure 4 is better to be close to the text in Section 4.1. The position of Figure 8 can also be adjusted.

That is correct – the vertical dashed lines are the mean values, which is mentioned in the caption. We will add the requested values to the plots in both figure 4 and figure 5. I agree with the positioning, and we will do our best to keep figures near relevant text. However, this is something that will change in the copyediting process and is somewhat beyond our control.

L366. The exception is not only Ross Sea. Am-Bel is 4 cm thicker in spring than summer.

Thanks for pointing this out – we will update the text to this point. The other reviewer also had comments on this figure, so this section – including the sentence you mentioned – will be modified in the revised manuscript.

L384. The last word "that" is 'than'?

This will be changed.

L395. "thinner that" to "thinner than"

We will make this change in the revised manuscript. Thanks for catching these!

P21. Figure 9. Use different colors to differentiate decreasing and increasing regions instead of the uniform grey?

Good idea – these were meant to just signal statistically significant trends, but I like the suggestion of coloring by their region-average trend. We will add in these colorings in the revised manuscript provided that the figure remains in the revised version.

L476. About the discussion of the radar-laser difference in this section, another possible reason may be the high spatial resolution of ICESat/ICESat-2 than the grid/segment-averaged coarse resolution CryoSat-2. The lower resolution may smooth higher or lower signals values.
Yes, this is a valid point and was not adequately mentioned in the original version. We will add in some information regarding the resolution differences between the two satellites. (Additionally, this difference was explored further in a previous work - Fons et al.2021; Earth and Space Science -  so we will also add in a reference to that work as well.)

P25. The legend in Figure 10 is "Sea ice thickness derived with CryoSat-2 snow depth" instead of "CryoSat2 snow depth"?
This will be changed – thanks.

L701-703. The citation of Meredith 2019 can be improved according to their suggestions: https://www.cambridge.org/core/books/ocean-and-cryosphere-in-a-changing-climate/polarregions/8D76B8865B796C16991F7A9FB6271C2D.
Thanks for pointing this out. As they recommended, the new citation will be changed to: (Intergovernmental Panel on Climate Change (IPCC), 2022).

---

## Author Response (AR1)

Response to Anonymous Referee #1

Summary:

This paper exploits an innovative method to use CryoSat-2 radar altimeter observations to retrieve first freeboard and snow thickness on sea ice before using both to estimate the sea ice thickness and volume - including a credible estimation of the uncertainty. The method has been developed and published in different publications and is therefore not described in-depth in this contribution. Here the focus lies in the illustration and discussion of pan-Antarctic and regional distributions of sea ice thickness and sea ice volume as based on 11 years of CryoSat-2 data - including a trend analysis.
Certainly this is an interesting and also important piece of work which broadens our knowledge about the thickness and volume distribution of Antarctic sea ice.

Dear Reviewer,

Thank you so much for your thorough comments on this manuscript. We appreciate the time and detail that you put in to helping to improve this study.
Below, you will find our responses (blue) to your comments (black). We agree with many of your points, and made the changes to the manuscript that are outlined below.

Thanks again,
Steven Fons, Nathan Kurtz, and Marco Bagnardi

General Comments:

GC1: You dedicate quite some part of your paper to a trend analysis. The relevance of this trend analysis is not sufficiently well motivated and not put into a credible context with the overall variability of both the Antarctic sea ice cover and its influencing factors. The added value of this analysis is not fully convincing. See my specific comments in this regard.
Thanks for this point. We agree that a good amount of this paper was dedicated to trend analysis. Our thoughts were that in presenting a long time series of CryoSat-2 estimates, we can help to provide some information on trends from a single satellite, unlike what has been done before using multiple satellites. This avoids the inter-mission biases and the associated uncertainty in previously-published trends.
With that said, we do understand that stressing the trends as we did with only 11 years of data and not fully acknowledging the multi-decadal oscillations can be problematic. Therefore, we have significantly toned down the discussion of trends in the revised manuscript. Instead, we presented a shortened section on "intra-decadal changes in the ice cover" (Section 4.5) and showed how the ice pack has changed in the last 11 years, no mention of the significance of trends. While we kept the term "trends" in (and also the objective metrics assessing the trends), I hope you will agree that we included plenty of caveats and explanations that these

are simply the observed changes of this period. In its place, more of the discussion section was devoted to fully discussing the comparison datasets and the limitations therein (see GC2).

GC2: Your paper contains only few elements of inter-comparing your product(s) with other, independent results. Here I feel your paper has substantial potential for improvement. On the one hand, the discussion included so far in the paper based on the comparisons carried out would strongly benefit from a more critical view of i) the limitations of the intercomparison data sets used and ii) a more careful investigation and discrimination of level versus deformed sea ice and/or mean versus modal sea ice thickness values. On the other hand, key intercomparison data sets are left out, kind of limiting the credibility of the results presented - especially when keeping in mind that the authors' estimation of freeboard, snow thickness and sea ice thickness are not independent and therefore require an even more careful evaluation. See my specific comments for more information.

We definitely understand this point, but also feel that there is a general lack of reliable datasets to which these data can be compared against. The Operation IceBridge (OIB) data products, for example, do not include any snow depth or thickness estimates from CryoSat-2 underflights in the Southern Ocean. However, we do understand the importance of including more comparisons, and have included a snow freeboard comparison with OIB (which **is** included for one flight in 2010) and a comparison to the extended ASPeCt data from Kern (2020).
With regards to your point on mean vs. model values, we completely agree and made it more clear that we are focused on mean (total level+ridged) ice only.
As mentioned in GC1, we used discussion section to discuss the comparisons with independent datasets and their limitations. This section includes the comparison to the extended ASPeCt dataset from Kern (2020) and discussion on these limitations.

Specific Comments:

Abstract:To my opinion, the abstract should contain a bit more information about the method, the product and its evaluation and less detailed information about the trend analysis results - simply because this is a short time series in a highly variable environment, possibly requiring 30+ years to derive any reliable trend information. See also GC1.
Fair point. We have substantially toned down the section on trend-analysis (GC1), and have updated the abstract accordingly in the revised manuscript.

L18: All three references given relate to the Arctic. I have i) difficulties to understand your choice to not directly focus on Antarctic conditions - as this is the focus of your paper and it does not read well to introduce / motivate an Antarctic focus paper with exclusively Arctic focus referenes - and ii) even for the Arctic the selection of the references given seems rather arbitrary, missing out several of the more recent literature that is available. I recommend to revise the references. Thanks for the comment. This section is intended to introduce the importance of sea ice thickness as a whole (for things like maritime navigation and indigenous

communities, which has historically been more pertinent in the Arctic), but I do see the point about focusing on the Antarctic right away. We have modified this sentence to read :
*Knowledge of sea ice thickness has long been important in the polar regions – from early Antarctic explorers navigating in icy waters (Herdman 1959) to indigenous Arctic communities traveling and hunting on the frozen sea (Nichols et al. 2004) – and continues to be a focus today for maritime navigation and climate studies (Meredith et al. 2019).*

Figure 1: I know, this is just a schematic figure. However, it wrongly implies that the part of the sea ice underneath the water surface is as thick as the part of the sea ice above the water surface. In addition, the thickness of the snow load almost certainly would lead to flooding of the ice-snow interface. Therefore, for the sake of displaying a more realistic schematic figure - that even lecturers might want to take from your paper - I recommend to replace this figure by one which has more realistic dimensions.
This has updated in the revised manuscript to the following:

[Figure]

In the caption, h_fs is not mentioned yet.
The term h_fs is mentioned in the caption: "…including snow depth (hs), ice freeboard (hf_i), snow freeboard (hf_s), ice thickness (hi)…"

L71: "... these estimates are both assumed to be biased high ..." --> There has been a Cryosphere Discussion paper around for a while (tc-2021-227 by Wang et al.); if I recall correctly they took an independent look (not from the producer's side) at the ESA sea ice thickness product. It might be worth a look.
Thanks for the reference. The Wang et al. work focuses on ICESat and Envisat, while the statement in the manuscript discusses the high bias found in the CryoSat-2 results only.

L121-125: ICESat had only several dedicated measurement periods while ICESat-2 has been operated continuously. I therefore assume the climatology maps have a different number of months as their baseline, i.e. for August or December it is possibly mostly ICESat-2 - aka data from 1 or 2 years, respectively, while for March it is data from one ICESat-2 year and five ICESat years. I recommend to include a short table detailing this difference in representativity of the climatology freeboard maps of the different months.

Not clear as well is how the different coverage of ICESat measurements over different months is taken into account in the respective monthly mean. Often these maps are bi-monthly maps derived e.g. half from February and half from March. How is this realized in your climatology? Did you use a Feb/Mar ICESat map for both February and March?

Good suggestion about the table. We have added this to Supplement, Table S1.
While you are correct that ICESat operated in discrete campaigns (and that the campaign files cross months), the full-resolution ICESat data still contain time stamps, and therefore only data collected in a given month is included in the initialization map for that month.

L131: "area of the grid cell" --> please provide the information where you obtained the grid cell areas from. Since this data is on a polar-stereographic projection the grid cell area varies with latitude and you possibly downloaded and used the respective file from NSIDC (?)

Thanks for bringing this up, and that is correct. Grid cell areas were computed with the NSIDC files (NSIDC-0771), which was not adequately described in the original manuscript. We have added this into the section on datasets and referenced this NSIDC product.

L137-139: Undoubtly the Worby et al. (2008) data set is a benchmark in this direction. I note, however, that it terminates in March 2005. Have you considered to take a look at the extension of this data set available here: https://www.cen.uni-hamburg.de/en/icdc/data/cryosphere/seaiceparameter-shipobs.html ?

Thanks for the suggestion. We did include a comparison to the updated ASPeCt dataset from Kern (2020) in Section 5 alongside a discussion on caveats of comparison datasets.

Table 1: In the text you state "angular backscatter efficiency"; I suggest to use the same expression in the table.
I note that snow depth seems to be given in cm while the roughness has a different unit. You could consider harmonizing this.
I am a bit puzzled about the bounds. For sigma you state bounds 0-1m; I assume this means that sigma is allowed to range between 0 and 1 m. However, for snow depth you specify a plus/minus range around the values suggested by the climatology rather than a range such as specified for sigma - otherwise the snow depth would need to range between -30 cm and +30 cm. Even when it is the range around the climatology values (which I assume) I am wondering what happens at a snow depth of 5 cm.
What is "std"?
I note that the static parameters don't have bounds even though you apply the retrieval year-round and backscatter / extinction characteristics of snow and ice may change throughout the year. Would it therefore make sense to introduce bounds here as well?

Thanks for the comments. Our responses:
- Thanks, we have updated to "angular backscatter efficiency" in the revised version.
- The units have been harmonized in the revised version as well.
- For sigma, it is always initialized to the same value, and we provide bounds that are expected over sea ice (0-1m). The snow depth input varies, and therefore we can't assign a set range. Instead, (as you correctly guessed) plus/minus around the input value is used. However, we did neglect to mention that if the input is below 0.3 m, then the

lower bound is 0 (i.e. does not go negative). We added in a sentence better explaining this.

- Std = standard deviation (to differentiate from sigma already used). We changed this to "SD" for consistency and defined it in the caption.

- These parameters are fully static, and therefore do no change. So it wouldn't make sense to introduce bounds. While we would ideally have these as dynamic parameters and vary them by season/region with bounds, there is not enough information to accurately constrain them, or know how they vary throughout the year. We explored this further in (Fons et al. 2021, https://doi.org/10.1029/2021EA001728), but have added this into the revised manuscript, L187.

L175: "or until … is reached" --> What are the output parameters taken in this case? The very last one? Also, I asssume that finding a minimum residual results in a quantitatively better fit. How often does the retrieval needs to reach 100 evaluations compared to finding an adequate minimum?

Good catch, and apologies for not including this. In the event there is no convergence, the waveform is discarded. This happens very infrequently (< 0.5%). More often, a minimum solution is reached, but is a poor fit (with a large residual). The waveform will then be discarded by the filtering process. We have added this information in the revised version, as well as info on the total waveform discard percentages per the other reviewer (~14%).

L183: "the nominal tracking bins" … what is their difference and hence an approximation of the vertical resolution of the approach?

The nominal tracking bin is a value provided in the CryoSat-2 data products. This is the location (in range bins) to which the range from the satellite to the surface is computed, and from which the retracking calculation is obtained. This clarification has been added to the revised manuscript.

L207: "if at least three lead-type points exists within" --> I am sorry for asking this question but can successive points overlap or are they truly independent, i.e. adjacent footprints do not overlap because the along-track distance between their centers is larger than the along-track dimension of the footprint?

No problem, thanks for bringing it up. While each CryoSat-2 shot does overlap in the along-track direction (due to the large footprint), the SAR processing of the waveforms keeps the measurements fairly "independent" in the along-track direction by slicing the measurements up into strips that are fairly narrow along-track. Due to this processing, we can assume successive points are independent enough for the purposes of finding candidates SSH points.

L237: I could have asked this question earlier in the context of the SSH approximation: For that approximation you need a minimum of 3 valid points within a 10-km segment. And then you first compute the parameters mentioned along track, i.e. for each valid floe-type point along-track, and then perform the gridding?! For the latter, does that have to be a minimum of valid floe-type points from which the parameters mentioned are computed? I can imagine that there

are seasons and regions where you may have quite a number of valid lead-type points and a lot of mixed-type points but only few floe-type points.

Finally, the SSH derived is representative for 10-km segments, i.e. in the worst case a step-function in SSH, or is this derived using a running 10-km segment possibly providing a smoother representation of the SSH?

This is a good point to bring up. The parameters are computed for each waveform, so there is no minimum amount in that case. For the gridding, we neglected to mention this (sorry), but we use 5 waveforms as a minimum number of waveforms to create a grid (following Kurtz et al. 2013). We have added this in the revised version, L240.

Also, the 10-km SSH is done using a running segment, since you are correct that discrete 10km segments would create a step function SSH. We have added this clarification to the revised version, L231.

L278/279: While following the approach of Spreen et al. (2009) is at first place good, I am wondering whether it would make sense to back this value up by looking into sea-ice concentration uncertainty information that is provided with the OSI-450/OSI-430-b CDR/iCDR sea ice concentration data set (which includes smearing uncertainty contributions) or with the NOAA/NSIDC SIC CDR (even though this is basically a modified standard deviation)? Another source you could look into in this regard is this one:

https://egusphere.copernicus.org/preprints/2022/egusphere-2022-1189/

Thanks for the comments. While we acknowledge that these methods would improve the sea ice concentration uncertainty estimate, we stress in the paper that this is a simple approach used to provide some idea of the volume uncertainty, and feel that the conservative estimate of 5% works for this purpose. Future work that is focused on more fully constraining the uncertainties in sea ice thickness/volume retrievals would benefit from these studies.

L310: "more validation data" --> I would call your comparison to the KK20 product an inter-comparison. It is not a validation as the KK20 data is just one possible result of combining two satellite data sets to obtain a snow thickness product. For an evaluation or validation one would need ground-truth data which the KK20 data certainly is not. Hence, my suggestion is to stress that we need "validation data" (hence delete "more" as this implies that the KK20 data are already validation data) of the type ground-based measurements to really be in the position to perform a validation.

Very good point, thanks. We have made this change in the revised version.

Figure 3: I am wondering whether it would make sense to provide an estimate of the actual number of values per month as I would assume that the number grid cells contributing to a February value in the Indian Ocean sector differs considerably from the region W-Weddell.

I note that the name for sector "Amundsen-Bellingshausen Sea" has different flavors. Consider using one.

Yes, it probably makes sense to include this information. However, this figure is mostly a comparison to KK20, and therefore both sensors would experience similar relative amounts of measurements in each season. We have added in some text acknowledging that comparing regions must take into account the different number of values per month, L355.

Good catch with the "Amundsen-Bellingshausen". We have ensured the revised manuscript is consistent, sticking with "Amundsen-Bellingshausen Seas" as the full name, abbreviated to "Am-Bel" in most of the text.

L334-338: "In Fig. 4 ... in Fig. 3)." --> How do these results compare to the snow thickness retrievals based on ICESat data in Kern and Ozsoy-Cicek (2016)? Didn't they also show an increase in the average snow thickness from autumn to winter to spring - in contrast to the snow thickness values derived using a modified version of the Markus and Cavalieri (1998, online 2013) approach?

Kern and Ozsoy-Cicek (2016) only looked at changes from Winter to Spring (and comparisons showing thicker spring snow depths than AMSR-E), but you're right that this would be useful here (and both studies found similar results). We have added in this reference and our broad-scale comparisons to that work in the revised version.

Figure 4: The dashed line in the inset histogram denotes what?
The legend of the maps as well as the histograms is in meters. I suggest to then also set the binsize to 0.02 m.

The dashed line denotes the histogram mean (text added in the revised version). And good point about the units. We feel 2cm is easier to understand that 0.02m in a figure caption, but also get the need for consistency and so have updated this in the revised version.

L355 / Figure 6: I am wondering about the real information content of these probability distribution values given the fact that the number of observations per region / season varies so much. Did you consider normalizing the histograms to 1?

The histograms are currently normalized so that the area of the histogram integrates to one, which tends to be a more robust normalization technique.

Figure 5: Same comments as I had for Figure 4.
In addition: please swap "Spring" and "Autumn" below the bottommost row of panels.

Good catch! Thank you. These have been swapped in the revised version, and the dashed-line explanation was added to the figure caption.

L363-368: "Despite ..." --> I have repeatedly used the Worby et al. (2008) data set (and its extension mentioned further up in my comments) for inter-comparison purposes and am well aware of its value. I am wondering, however, whether some additional information needs to be given here to underline how vague that information can be. These data have an observational negative bias because ships tend to avoid thicker sea ice. In summer, floes break different under the action of the ship's hull reducing integrity of level ice made of rafted ice floes. In summer, pancake ice which makes a substantial fraction of the observed sea ice cover, is essentially lacking. Leads, often followed by the ships, are covered by thinner ice types in summer than the freezing season. Also, in addition to these observational biases there could be biases by the regions traversed during the different seasons. For instance in the Ross sector, cruises hardly reached to the thicker sea ice parts in the eastern Ross Sea during winter and spring, simply because these areas are not accessible, but rather crossed the thinner sea ice in

the sea ice export area of the Ross Ice Shelf polynya. Therefore, especially the high sea ice thickness reported in Worby et al. (2008) for summer could very well be caused by preferably entering areas with thicker sea ice compared to winter and spring. See GC2.

Thanks for this additional information, and you're exactly right that there is some additional info that could be added to this manuscript. The thought was provide this comparison and stop short of any robust intercomparison, mainly due to these caveats in the ASPECT observations. However, we do see the utility in adding in this information. We have substantially reworked the discussion section and included more information about the comparison datasets and the limitations of each of them, Section 5.1.

Figure 6: Are the values from Worby et al. those of the level ice or do these include the estimated contribution from ridged sea ice?

These are the average ice thickness values, so they include contributions due to ridging. They are compared to the mean of the CryoSat-2 data, which includes the ridged ice (as opposed to the modal thickness values).

Figure 7: While I was trying to understand why I have the impression that the individual mean sea ice thickness values do not add up to the pan-Antarctic mean sea ice thickness value I figured out that the scales are not the same. How important is it (for your message) to show the pan-Antarctic sea ice thickness occupying more vertical space in the figure than the individual sea ice thickness time series? Would it make sense to try to show the time series for each region with the same vertical scale?

Yes, apologies for this. We had originally made the plot with identical axes (and gone back and forth about which to include), however, you lose a lot of detail in the individual region plots, as the range can be large on some and small on the others. In the revised manuscript, we have reduced the size of the pan-Antarctic subplot and made the scales the same (except for the pan-Antarctic, which is zoomed to better show the annual cycle) to make it easier to interpret this figure.

L390/391: "while Maksym ... thickness" --> This sounds like they used satellite microwave radiometry to estimate sea ice thickness but what they did is first of all not that simple and secondly their main statement refers to level, undeformed sea ice which is not 1-to-1 comparable to your work. I therefore invite you to check the reference one more time and to rephrase your sentence accordingly. It is important to check out which part of the sea-ice thickness distribution the respective publications refer to to be able to make appropriate statements here. In this context it might be a good idea to, in addition, introduce a discussion of modal sea ice thickness values representative of the level sea ice.

Fair point, thanks for bringing it up. We did lump these all together, when we should have added in more information on the mean vs. modal thickness from each study. We have revised this section to better point out the caveats of each of these "thin" estimates and why that may be the case. See revised L438-456.

L398/399: "Williams et al. ..." --> You might find it enlightening to again take a look into Kern et al. (2016). Even though their 1-layer method results are possibly biased and rather refer to the

total (sea ice plus snow) than the "true" sea ice thickness, the intercomparison of the other methods (including the ZIF) seems to provide a possible range (at least for the ICESat measurement period) of sea-ice thickness values obtained using different methods.

Good point – Kern et al. does show examples of estimates that are indeed thicker than other observations, and could provide some argument for a 'thicker-than-expected' ice thickness. We won't try to directly compare to these results (due to the different time periods), but have added in a refence to this work to L446.

Section 4.4: In light of the substantially larger (and known) variation of the Antarctic sea ice cover - compared to the Arctic - I have a conceptual problem with dedicating a full sub-section to a trend analysis of an eleven years long time series. This looks like somebody wants to investigate an eleven years long precipitation time series of the U.K. in light of trends. But it is of course your decision to keep or delete this part of the manuscript. In case you keep it I strongly recommend to - beyond statistical significance estimates - state clearly that any trend found for these eleven years can simply be the part of a multi-decadal variation that cannot be resolved yet with the existing record of CrysoSat-2 sea ice thickness and volume observations. This would be a good motivation to i) discuss your results even more in the context of the work of other studies; ii) to advertize more work needs to be done to include Envisat and ERS1/2 RA altimeter data analysis to extent the time-series; iii) to advertize your own sub-section about expanding the CS-2 time series back in time to the ICESat periods.

This is a very valid point, this section has been reworked in the revised manuscript (See GC1).

L420: "contains at least four years of data" --> Is there any constraint as to when these four years need to contain data? Is it possible that all data are from the first 4 years?

There is no constraint applied, other than having a minimum amount of data. This really only impacts the ice margin, where the extent changes more drastically year-to-year, and does not impact the "central" ice pack nearly as much. We have added in a sentence mentioning this caveat to L479.

L432: "Holland (2014)" --> How many years of your 11-year period overlap with the data used by Holland (2014)? Are those results therefore compatible with your results?

There is not much overlap between the two datasets – just part of the year 2010. This is why there is not much discussion given to this comparison (also because of the differences inherent in a modeling study vs. observations). We have made this clear in the revised version that the time periods share little overlap.

L432/434: Both, Garnier et al., (2022) and Xu et al. (2021) used data from a longer time series, didn't they? What is then the added value of performing such a trend analysis over a shorter time period? This is not entirely clear to me.

It was mainly done to showcase the trends one can retrieve from a single sensor. All other studies either combined data from multiple sensors (the ones you listed), or showed a time series from a single sensor that was shorter than ours. The revised manuscript mentions these only briefly in the modified section on intra-decadal changes in thickness.

L441-443: How many years of CryoSat-2 data did Kwok and Cunningham (2015) use in their analysis? I checked it out: It is four winters. You investigate 11 years. I don't think your current writing (and citing that paper) does support further discussing the impact of an analysis of 11-years worth of sea ice thickness and volume in the Southern Ocean.
See GC1 – the trend analysis section has been substantially revised in the revised version.

Below in this sub-section you will find more comments going into this direction. All I wish to trigger with these is to encourage you to one more time critically think whether the message you provide here is compelling, sustainable and worth the effort. Does it send out the right signal in view of already existing work and in view of what we know about the length time series of geophysical parameters should have in order to provide a meaningful statement about climatological features such as trends? See GC1.
See GC1 – the trend analysis section has been substantially revised in the revised version.

L455-457: "However, the same ... since 2014" --> Certainly. And if you shorten the time period even further, e.g. to a 4-years like Kwok and Cunningham did, then you will find an even larger decrease in sea ice thickness or volume for 2014-2017 while you may find an increase in sea ice thickness or volume for 2011-2014 and 2017-2020. Fine. And?
See GC1 – the trend analysis section has been substantially revised in the revised version.

I find it kind of dangerous to refine the temporal granularity of such trend analysis in an area such as the Southern Ocean being influenced by at least three multi-decadal oscillations plus El Nino/La Nina events. I agree, Kwok and Cunningham (2015) did it with an even shorther time series, Kurtz and Markus (2012) as well ... but what did we learn from these?
See GC1 – the trend analysis section has been substantially revised in the revised version.

L463/464: "modeled studies into ... scenarios" --> Certainly. But this is not a surprizing finding and, in addition, it requires first some more work still to be done improving those models - see Roach et al., 2020, Geophys. Res. Lett., 47, who for good reason first looked at the Antarctic sea ice area in CMIP6 models finding it not well represented.
This section was removed in the revised version.

L472/473: "A longer-term time series ... implications." --> Exactly. Two other studies exist (almost certainly there are more in the meantime) that already looked into longer time series which complicates to see the immediate added value of your investigation in comparison to their studies.
Se See GC1 – the trend analysis section has been substantially revised in the revised version.

L497-501: "Likely ... estimate Antarctic sea ice thickness." --> I have two comments here. The first one is related to whether you also looked into the work of Ricker et al., 2015, Impact of snow accumulation on CryoSat-2 range retrievals over Arctic sea ice: An observational approach with buoy data, Geophys. Res. Lett., 42. While being for Arctic conditions that work might be further enlightening with respect to your observations.

Yes, we are familiar with this work, which could perhaps inform what we're seeing here. However, we feel that there is plenty of evidence for overestimation of sea ice thickness in the Antarctic using the threshold retracking method (works cited in text) and that tying in more Arctic work would overcomplicate things in this section.

The second comment is about the observation that in the time series of h_i_70 the primary maximum mean sea ice thickness is not occurring in February anymore but occurs in late winter / spring in all but one year. What does this tell us in light of the fact that the primary maximum now occurs close to the maximum sea ice coverage - involving a large fraction of seasonal sea ice with different surface properties than encountered in February?
This is a very good point, and one that was not discussed in the original manuscript version. It could be a number of reasons, one of which ties in to your above point of snow accumulation. Since this method is doesn't explicitly account for the impacts of increasing snow load, it could retrieve an increasingly anomalous thickness snow accumulates throughout the season. We have added in some discussion on this differing yearly maximum in the revised version, Lines 582-591.

L505/506: "could come from ... its lifetime" --> I am aware of these changes but at the same time I am wondering i) which release of the ICESat GLAS data you used for your re-processing of the ZIF sea ice thickness values and ii) whether you did not correct for the different gain values that are reported along with the ICESat data?
These data are taken from https://earth.gsfc.nasa.gov/cryo/data/antarctic-sea-ice-thickness, using the method described in Kurtz and Markus (2012), and are not corrected for gain values. However, they only use data through 2008 to avoid issues with instrument degradation.

L525 / Section 5:

I absolutely agree with you that it would be really nice to have ground-based observations that cover all three sensors' observation period. But we know that this is not possible. The only data sets I am aware of that covers all three sensors contain only estimates of the sea ice thickness: the ASPEcT data set and its extension mentioned further up. Arctic studies often tend to look into PIOMAS to see whether there is long-term consistency in the estimates. I am not deep enough involved into such studies to know whether GIOMAS data would be a viable alternative for the Southern Ocean.
However, apart from these considerations, I am missing a more thoughtful evaluation of your sea ice thickness data / product for the CryoSat-2 period used. I have several concerns. One is the apparent lack of adequately discriminating between modal (level) and mean (level + deformed) sea ice thickness values in those parts of your intercomparisons where such a discrimination would be possible (e.g. the Worby et al., 2008 data). In that context I note again that you could have used the extended version of these data noted earlier in addition - even though these do not contain this discrimination into level and level+deformed ice.

In this context I would like to remind you to adequately discuss the limitations of the data you

used for your intercomparisons presented in this manuscript - as voiced further up in the context of the Worby et al. (2008) data.

Thanks, a section of discussion on these limitations has been added to the revised version, see the GCs and revised Section 5.1.

What I am missing is consideration of Operation Ice Bridge data in your evaluation and discussion of the quality in this manuscript. There is a substantial amount of data available and even though flights mostly cover the Weddell and Bellingshausen Seas these are nevertheless a very valuable source for the evaluation of your product. Other air-borne data exist, such as helicopter-borne electromagnetic sounding but I am in fact not sure how many of these would be available within the CryoSat-2 period. For sure researchers organized from New Zealand obtained data in the southern Ross Sea.

While Operation IceBridge did indeed collect a substantial amount of measurements over Antarctic sea ice, what we are missing is a sea ice freeboard/snow depth/ thickness product for these data. There are only 6 flights in which freeboard data are provided by NSIDC, covering October 2009 and 2010 (3 during CryoSat-2 period). Of these, only one performed an underflight of CryoSat-2, however, it only provides snow freeboard (no snow depth nor thickness estimates). While one could generate their own product, it would require a separate study to validate and trust the results.

Nevertheless, the other reviewer also suggested further comparison, and this underflight would be a good addition. In the revised version, we have provided a comparison to OIB snow freeboard from 28 October 2010 in the Weddell Sea, revised Figure 3.

In short, in view of recommendations I conveyed to other authors with a similar manuscript profile my main recommendation for you and your section 5 is to put more emphasis on more critically discussing the reliability of your results rather than discussing trends.

We appreciate the comments, and have substantially reworked the discussion of trends and reliability of results in the revised version. See GC1/2 for more information.

Editoral Comments / Typos:

L25: "snow freeboard" --> You could add that here the assumption is that the dominant scattering comes from the snow surface.

Good point. This does help with the clarification and has been added in the revised manuscript

Equation 2: I recommend to add the information that the second term actually results in a reduction of the sea ice thickness computed by the first term alone - which is opposite to Equation 1 - and which particularly in the Antarctic - the focus of your paper - is important to consider as snow freeboard might equal the snow thickness or may even be smaller than that in case of flooding.

This is a very good point. The following sentence has been added:

"It is important to note that in Eq. (2), the second term results in a reduction of the sea ice thickness computed from the first term alone, which is opposite that of Eq. (1) and can play a

key role for Antarctic sea ice where snow freeboard may equal (or be less than, in the case of flooding) the snow depth".

"Kurtz and Markus, 2012" and "Kwok, 2011" are references in which one can find these two equations - however, I am wondering whether it wouldn't make more sense to go back to those publications where these equations were developed / introduced first ... which might be the Laxon et al. paper from 2003 in case of Equation 1 and one of the earlier Kwok (et al.) papers for Equation 2.
Good point, and thanks for the suggestion. It appears to be Laxon et al. 2003 for equation 1 and Zwally et al. 2008 for equation 2. We have added these in the revised version.

L36: ICESat facilitated "snow freeboard" measurements. Please correct.
This "sea ice freeboard" was meant more in the general "freeboard of sea ice", but I do see the confusion. We have corrected this in the revised version.

L39-42: "In most of these ... Kurtz et al., 2009)" --> I suggest to place the Warren et al. reference behind "1954-1991"; otherwise it reads as if Warren et al. (1999) have used that climatology to convert freeboard to thickness.
Good point, thanks. This has been modified in the revised version.

I further suggest to not highlight that Kurtz et al. (2009) used snow thickness data from passive microwave sensors (which by the way do not provide "lower resolution" snow thickness data compared to the Warren et al climatology being based on interpolation using a polynomial function anyways) - simply because this is just one of the alternatives used by the various other groups already cited. How important it is for the Antarctic focus of your paper to introduce the reader to potential alternatives to the Warren et al. climatology which is not existing in the Antarctic?
The 'lower resolution' was not meant to be a comparison to the Warren Climatology, but instead a general statement on low-resolution snow depth data. We do see how this is confusing and have modified this in the revised version.
We do feel it is important to bring up all the various snow depth data used in the Arctic, as a way to drive home the contrast between the available data in the two hemispheres. That said, it may not add much to the overall message, and can instead highlight in our revised version that snow models are used more frequently (recently) in the Arctic. See L47.

L43: ICESat-2 --> Did you overlook the contributions by Kwok (et al.) - who also combined Cryosat-2 and ICESat-2 - on purpose here?
This section is on studies retrieving freeboard/thickness from a single sensor, either CryoSat-2 or ICESat or ICESat-2. Kwok et al. 2020 uses combined ICESat-2 and CryoSat-2, which is the topic of a later paragraph, found on L91.

L45: "have found success in estimating sea ice freeboard over Arctic sea ice" --> In light of the fact that most of the studies you cited had freeboard-to-thickness conversion as their ultimate

aim, I am wondering whether you might want to rephrase this along the lines: "were succesful in retrieving sea ice thickness from sea ice freeboard estimates over Arctic sea ice" ... or the like.

Good suggestion, thanks. That does make more sense given the focus of this work, and has been changed in the revised version.

L57: "Markus and Cavalieri, 2013" --> This is the electronic version of the original book chapter from 1998, right? Has the content changed? If not, please check with EGUSphere how to cite to avoid the impression that this is a more recent work.

Apologies, and thanks for catching this. There was an error in the .bib file that showed an incorrect year. The correct year has been cited here in the revised version.

L74: "through the use of key snow depth assumptions" --> I might be wrong but it is only the Kurtz and Markus (2012) work which does this assumption. I therefore suggest to add something like "partly" or "for example" to make clear that assuming zero freeboad is ONE possible solution - with limited applicability though as one can figure out in the subsequently cited by you literature.

Done.

L78: "Zero ice freeboard ..." --> In addition to citing Willatt et al (2010) you could also include Ozsoy-Cicek et al. 2013, JGR-Oceans.

Good suggestion – this has been added.

L79: Regarding this underestimation you could have cited the earlier study by Kwok and Maksym from 2014 (JGR-Oceans) using OIB data; also Kern et al. (2016) performed intercomparisons between different retrieval approches, Kurtz and Markus being on of these.

Thanks for the suggestion. Citing these works here is indeed a good idea, and has been done in the revised manuscript.

L95 "utilize CryoSat-2" --> It would not hurt to also mention Envisat here because with that one would have an uninterrupted time series produced using an independent sensor from 2003 through today (see also Paul et al., 2018).

Good suggestion – we have mentioned Envisat here in the revised manuscript, and still highlight the importance for having CryoSat-2 to fill the gap.

L127: "is based off of" --> I would have written "is based on" ... but I am not a native English speaker ...

Yes, I believe you are right. Thanks for catching it – we have made the change in text.

L160: You might want to change the font of P, I and p so that it matches "v" and equation (4).
L187: "R_n" needs to be "R_0" ?

This has changed, thanks.

L197: I am not sure I would throw the Schwegmann et al. paper into one pot with the Paul et al one because the latter used a considerably modified methodology. Hence citing Paul et al might be sufficient here.

Fair point, thanks. We have removed the Schwegmann reference here in the revised manuscript.

L239-241: You could consider to delete the information about which algorithm you used and how you compute the sea ice area because you described this earlier.

Good point, thanks. This has been removed in the revised version of the manuscript.

L247: You might want to add that the demarcation in longitude is given in the unit "degrees East".

Thanks for the suggestion – I agree, and have make that change in the revised manuscript.

L262: The "s" in h_fs needs to be put in sub-script mode.

This has been changed in the revised manuscript.

L267: "h_f" at the end of the line needs to be "h_fs"?

This has been changed, thanks.

L293: "sections" --> "section"

This has been changed, thanks.

L301: What is "IQR"?

This is the inter-quartile range, which (we realize) was defined in the figure caption but not in the text. We have changed this to be "Inter-quartile range".

L308/309: "Despite ... nevertheless" ... I guess one of these is enough; I'd discard the "nevertheless".

Fair point, thanks. We removed "nevertheless".

L325/326: "This could cause ... anomalous snow depths" --> I am wondering whether you could narrow this down towards that the snow-ice interface will most likely be located higher in the snow pack and then also state that this will lead to anomalously low snow thickness values?

Good point, this can definitely be clarified more along the lines of what you suggest. We did mean anomalously low snow thickness values, but (obviously) just left it at anomalous. This has been updated in the revised version.

L379/380: You are refering to "basal growth" here. For Southern Ocean sea ice a substantial portion of the sea ice volume (up to 1/3 in some places) is actually made of snow ice, i.e. snow that was first flooded at the ice-snow interface and then re-froze. This is not a basal growth. One solution could be to simply write "growth".

Thanks for the suggestion – yes, we called it 'basal growth' but indeed did mean to include other types of growth processes as well. We changed this to 'growth' in the revised manuscript and referenced the possibility of growth through snow-ice formation.

L384: "in ice thickness" --> in pan-Antarctic sea ice thickness"
Good suggestion – this clarification has been added in the revised version.

"more that" --> "more than"
This has been corrected.

L394: A reader would be happy to be reminded what this correction factor does and when it is applied.
Yes, agreed. We included a note about what the correction factor is and how it was used in that study to adjust the snow depth due to displacement of the scattering surface.

L395: "was estimates" --> "was estimated"
This has been corrected.

L459: Sometime you use pan-Antarctic with a capital "P" sometimes not. You might decide for one version of how to write it. I don't know actually what would be correct grammatically.
We've confirmed that it should be written with a lowercase "p", and have made that change throughout.

L484: "about" --> "around"
This has been changed.

L510: "h_i-total" is what?
This is the thickness of ice derived from the total freeboard. We realize this was defined as $h_{i-sfb}$ in equation 2, so made the change here. Additionally, we harmonized the use of $h_{i-ZIF}$ and $h_{i-oifb}$ in this section with equations 1-3 in the revised version. They are now: $h_{i-fs}$, $h_{i-fi}$, and $h_{i-ZIF}$ for thickness derived from snow freeboard, ice freeboard, and using the ZIF assumption, respectively.

L550/551: "It is clear that ... sea ice thickness," --> I encourage you to also include "snow thickness" here.
Very good suggestion, thanks. We included a mention of snow thickness here as well.

L611: You might want to replace this reference by the paper published in Earth and Space Science, 8(7), 2021 to have the link to the peer-reviewed version of your work.
Thanks for catching that, it is indeed an old link that was carried through in the .bib file. We updated this link to the final, peer-reviewed version of that manuscript.

Response to Anonymous Referee #2

This paper presents an important work to not only fill in the gaps of the ICESat/ICESat-2 observations on the Antarctica sea ice thickness, but also provide a continuous ice thickness time series showing obvious seasonal cycle characteristics with CryoSat-2 from 2010 to 2021. The authors utilize a physical model and a waveform fitting method that they developed in their previous work to get snow depth and total freeboard, then the sea ice thickness and volume. This work provides a sea ice thickness dataset that could be merged with that derived with ICESat/ICESat-2 to produce a longer-term observations of circum-Antarctica sea ice, which would greatly promote global climate change studies. However, there are some concerns that need to be clarified by the authors before publication.

Dear Reviewer,

Thank you so much for your comments on this manuscript. We appreciate the time you put in as well as your contributions to improving this study.
Below, you will find our responses (blue) to your comments (black). We agree with many of your concerns, and made the changes to the manuscript that are outlined below.

Thanks again,
Steven Fons, Nathan Kurtz, and Marco Bagnardi

Major:
It is not clear at which step what parameters are estimated. Generally, there are several parameters involved here: total freeboard (air-snow interface elevation-derived), ice freeboard (snow-ice freeboard elevationderived), snow depth, and ice thickness. L181 states the ice freeboard and snow depth are output parameters produced directly by CS4WFA? However, L292-293 shows the snow depth are derived from subtracting snow-ice freeboard elevation from air-snow interface elevation. So, my question is what is the input and output between each step in this study? At which step comparing with what reference dataset? I suggest supplementing a flow chart to make it clearer.
Thanks for the suggestion, and apologies for the confusion. Some of these processes can be described in different ways. For example, the snow depth **is** an output parameter from the model, but is identical to subtracting the interface elevations and therefore can be "calculated" either way. Additionally, the snow-ice interface tracking point is output from the model, which is used to find the snow-ice interface elevation (via equation 5) and then ice freeboard. The updated better describes some of these processes in the locations you stated. In addition, we added the following flowchart as figure S1:

[Figure]

Where ovals show input/output data (green is start, red is end, white are inputs, grey is discarded), diamonds show decision/filtering processes, white squares show calculations/other processes, and blue squares show 'milestones' (inset orange squares for lead-type milestones, blue for floe-type milestones).

In Section 4, authors compare snow depth and thickness with other datasets. why not snow freeboard included in comparison?
This is a fair question. As mentioned in the text, a comprehensive snow freeboard comparison was done using ICESat-2 data in Fons et al. (2021; *Earth and Space Science*). We point readers there for this comparison, but I understand that it still feels lacking in this work. The other reviewer had similar thoughts, so we added in a snow freeboard comparison to Operation IceBridge data from a flight in the Weddell Sea in October 2010 to the revised version.

L179. About the statement 'Fit parameters are discarded if the result is a "poor fit"', how much data have been discarded and what kind of data are discarded?
Good question, this is info that should have been included. The number varies from orbit-to-orbit, but on average 86% of waveforms are kept (meaning 14% are filtered out). That number is higher for floe-type waveforms (21% filtered out). Only <1% of lead-type waveforms are filtered out. These numbers tell us that waveforms that are filtered out are typically "messy", meaning that have multiple peaks (due to scattering from off-nadir leads) and/or don't fit the typical profile of a return floe-type waveform. Lead waveforms are typically specular and without many off-nadir peaks, and therefore are less likely to be filtered.
We have added these values to the revised manuscript in this section.

The similar question for L213, for what types of waveform, there is no good fit? This is important because it may better inspect the proposed method.

Good question, with a similar answer to above. Poor-fitting waveforms typically have multiple peaks, usually brought on by off-nadir leads ( Kurtz et al. 2013, Tilling et al. 2018).  This is a common problem with many CryoSat-2 retracking methods, as the wide footprint is susceptible to returns from far off-nadir. We have added in this info to Section 3 of the revised manuscript.

Minor:
L1." a physical waveform model and a waveform-fitting method to estimate the snow depth and snow freeboard" is misleading. What is the relationship between the physical waveform model and the waveform-fitting method and are they used for snow depth and snow freeboard respectively or as a whole? Are they the comprehensive method called CS4WFA? The sentence is ambiguous.
You are right that they are all part of a comprehensive method that we're calling CS2WFA for simplicity. We construct a model and use the waveform-fitting approach to estimate output parameters, which includes snow depth and snow freeboard. We modified this in the revised version to: "We estimate the snow depth and snow freeboard of Antarctic sea ice using a comprehensive retrieval method (referred to as CryoSat-2 Waveform-Fitting for Antarctic sea ice, or CS2WFA) consisting of a physical waveform model and a waveform-fitting process that fits modeled waveforms to CryoSat-2 data. "

L3. Is the thickness in "snow depth and thickness" snow or sea ice thickness?
It is referencing the sea ice thickness; we added in "sea ice" here for more clarification.

L9-10. Some findings are expected after the sentence. For example, "…, showing the interannual differences between the two kinds of satellites". Or add "Results show that" before "Reconciling…"
Fair point. This has been changed to: "we place these thickness estimates in the context of a longer-term, snow-freeboard-derived, laser-radar sea ice thickness time series that began with ICESat and continues with ICESat-2, and contend that reconciling and validating this longer-term, multi-sensor time series will be important in better understanding changes in the Antarctic sea ice cover."

L80-82. There may be misstatements on those literatures:
The approach "Worby" in Kern 2016 is a static ratio between sea ice thickness and snow depth, which are seasonal empirical values from ASPeCt, which are ship-based observations.
Li 2018 is a dynamic ratio between snow depth and sea ice thickness, which are initial guess from empirical equations between the total freeboard and snow/ice thickness by Ozsoy-Cicek Burcu, et al; Sea ice thickness retrieval algorithms based on in situ surface elevation and thickness values for application to  altimetry;Journal of Geophysical Research: Oceans;2013, 118 (8):3807-3822. Xu 2022 is an improved version of Li 2018 with a similar strategy.
Wang 2022 uses the same 'Worby' method, i.e., the static ratio, in Kern 2016 for ICESat, while uses snow depth from AMSR-E/AMSR-2 for Envisat-based sea ice thickness retrieval.
So, I suggest the sentences to be rewritten. For example:
"With some empirical parameters, sea ice thickness can be estimated with ICESat/ICESat-2 alone. Regarding sea ice as a single ice/snow layer, the Worby method (Kern et al., 2016) uses a

static ice-snow ratio which are seasonal empirical values from the ASPeCt program (Worby et al., 2008). The one-layer method (OLM, Li et al., 2018) and its improved model (OLMi, Xu et al., 2021) are proposed with a dynamic ice-snow ratio for each footprint measurement based on an initial guess from the empirical relationship between snow depth/ice thickness and snow freeboard (Burcu, et al., 2013)."

Thanks for this suggestion. The idea was to categorize all of these based on their use of a snow depth/ice thickness ratio, but we do see how the way it is currently written simplifies the description of these studies. We also appreciate the suggestion, and have changed this sentence to:

"The Worby method (from Kern et al. 2016) used a static snow-ice ratio derived from seasonal empirical values from the ASPeCt program (Worby et al., 2008), while the one-layer method (OLM, Li et al. 2018) and its improved model (OLMi, Xu et al. 2021) use a dynamic snow-ice ratio for each footprint measurement based on an empirical relationship between snow depth/ice thickness and snow freeboard (Ozsoy-Cicek et al., 2013)."

P18. Figure 7. What caused the uncertainty difference between different sea sectors? How is the pan-Antarctic uncertainty computed?

The difference in uncertainty in the difference sectors comes from the difference in the snow freeboard, snow depth, and snow depth uncertainty between the sectors, since these are the variables in Equation 7 that vary spatially. The pan-Antarctic uncertainty is calculated simply as an average of the uncertainty of all grid cells basin-wide. This is a simplistic way to do this, but provides some estimate of where we expect the uncertainty to be. We added this caveat to the revised manuscript, and added that pan-Antarctic uncertainty is computed as an average of all grid cells.

L156. Add Ψ after "a physically-modeled waveform" for Eq.2
Good idea – this has been added.

L158. What is hsd in eq.4? If it is snow depth, should it be hs according to Eq.1 and Table 1?
That is correct, and has been changed in the revised version.

L187 Rn should be R0 here?
Yes, thanks for catching this. It should indeed be R0, and we have made that change in the revised version.

Title of Figure 2. For "the daily linear interpolation", to avoid misunderstanding that the dashed lines arederived with daily data, it may be better written as "the linearly interpolated dashed lines between the midpoint dates are used as daily density estimates for the thickness calculation."

Thanks for the suggestion - that is a small but important distinction. We have made these changes in the caption of Figure 2.

L262, hfs should be hfs. Is there system uncertainty for snow freeboard in Eq.7?
Thanks for pointing out the formatting error – we have made this change in Eq. 7.

Equation 7 is based off of Petty et al. (2020) and Ricker et al. (2014), and is a simplistic way to provide some measure of uncertainty. It does not include an estimate of systematic uncertainty for freeboard, since it is not reliably known. We added in a sentence here that we acknowledge this is a simplistic way to estimate uncertainty, and that the current equation is used until we have a better idea of the snow freeboard uncertainties.

L290. Section 4.4 missed in this sentence.
Thanks – we have updated this sentence to include a reference to section 4.4.

P15. Figure 4. Are the vertical dashed lines mean values? Can the mean, std, and model values in eachseason/month be put on the map? The same with Figure 5.
The position of figure 4 is better to be close to the text in Section 4.1. The position of Figure 8 can also be adjusted.
That is correct – the vertical dashed lines are the mean values, which is mentioned in the caption. We have added the requested values to the plots in both figure 4 and figure 5. We have also adjusted the positioning to be in a better spot relative to the text.

L366. The exception is not only Ross Sea. Am-Bel is 4 cm thicker in spring than summer.
Thanks for pointing this out – this section has been modified to:
"Additionally, all three find the Indian and Pacific regions to have the thickest sea ice in the summer, with Worby et al. (2008) also showing the thickest sea ice in summer in the Ross and Am–Bel regions. Both Worby et al. (2008) and Xu et al. (2021) find summer to be the thinnest season in the Western Weddell sector, matching what we show with CS2WFA."

L384. The last word "that" is 'than'?
Changed.

L395. "thinner that" to "thinner than"
Changed, thanks!

P21. Figure 9. Use different colors to differentiate decreasing and increasing regions instead of the uniform grey?
Good idea, however, we have removed the polygons marking regions of statistical significant trends, per the other reviewer's request. However, we kept this figure in to show regions of increasing and decreasing trends.

L476. About the discussion of the radar-laser difference in this section, another possible reason may be the high spatial resolution of ICESat/ICESat-2 than the grid/segment-averaged coarse resolution CryoSat-2. The lower resolution may smooth higher or lower signals values.
Yes, this is a valid point and was not adequately mentioned in the original version. We have added in some information regarding the resolution differences between the two satellites. (Additionally, we have added a reference to Fons et al.2021; Earth and Space Science who explored this idea in depth).

P25. The legend in Figure 10 is "Sea ice thickness derived with CryoSat-2 snow depth" instead of "CryoSat2 snow depth"?

Changed – thanks.

L701-703. The citation of Meredith 2019 can be improved according to their suggestions: https://www.cambridge.org/core/books/ocean-and-cryosphere-in-a-changing-climate/polarregions/8D76B8865B796C16991F7A9FB6271C2D.

Thanks for pointing this out. As they recommended, the new citation was changed to: (Intergovernmental Panel on Climate Change (IPCC), 2022).